# BiasBusters: Uncovering and Mitigating Tool Selection Bias in Large Language Models

**Thierry Blankenstein**[1], **Jialin Yu**[1,2,†], **Zixuan Li**[1], **Vassilis Plachouras**[2], **Sunando Sengupta**[2],
**Philip Torr**[1], **Yarin Gal**[1], **Alasdair Paren**[1], **Adel Bibi**[1,†]

[1]University of Oxford, Oxford, UK
[2]Microsoft, UK
[†]Corresponding authors

## Abstract

Agents backed by large language models (LLMs) increasingly rely on external tools drawn from marketplaces where multiple providers offer functionally equivalent options. This raises a critical fairness concern: *systematic bias in tool selection can degrade user experience and distort competition by privileging certain providers over others*. We introduce a benchmark of diverse tool categories, each containing multiple functionally equivalent tools, to systematically evaluate tool-selection bias. Using this benchmark, we evaluate seven LLMs and show that substantial bias persists, with models either fixating on a single provider or disproportionately favoring tools that appear earlier in the context. To uncover the sources of this behavior, we conduct controlled experiments that isolate the effects of tool features, exposed metadata (name, description, and parameters), and pre-training exposure. We find that (1) semantic alignment between user queries and tool metadata is the strongest driver of selection; (2) small perturbations to tool descriptions can significantly shift choices; and (3) repeated pre-training exposure to a single endpoint amplifies provider-level bias. Finally, we propose a lightweight mitigation strategy that first filters tools to a relevant subset and then samples uniformly, substantially reducing selection bias while maintaining strong task coverage. Our results highlight tool-selection bias as a key obstacle to the fair deployment of tool-augmented LLM agents. Our code and benchmark are publicly available at `https://github.com/thierry123454/tool-selection-bias`.

## 1 Introduction

Large language models (LLMs) have transformed natural language processing, achieving near-human performance on tasks ranging from code generation to creative writing (Naveed et al., 2024; Luo et al., 2024). Yet LLMs cannot directly act in the world: they cannot query databases, fetch live information, or invoke external services. Additionally, their knowledge remains frozen at training time, leaving them prone to "hallucinations" when asked about events beyond their cutoff (Ji et al., 2023). Augmenting LLMs with external "tools" / APIs addresses these shortcomings by allowing models to delegate specialized functions to dedicated services (Qu et al., 2025). It endows LLMs with the ability to act, a core capability often associated with LLM *agents* (Chowa et al., 2025). A crucial step within the typical tool-usage pipeline is the multi-stage tool-selection process: given an instruction to the LLM, (i) retrieve a short list of the most relevant candidate tools based on the user query (with, e.g., highest semantic similarity) from a potentially large database of tools, (ii) insert their metadata into the prompt, (iii) have the LLM reason and pick one to solve (one of) the necessary user task(s) (see Appendix K covering the entire tool-usage pipeline). However, this process introduces a new challenge: bias (see Figure 1). An LLM may prefer certain tools not for their relevance or accuracy, but because of superficial metadata, i.e., tool names, descriptions, or prompt ordering. Such bias can degrade user experience by repeatedly selecting slow or unreliable services and can, therefore, also inflate operational costs. Additionally, under pay-per-request pricing for tool use, consider the scenario where such biases are systematic across frontier LLMs: if they consistently favor tools from a single provider, this creates major market unfairness, disadvantag-

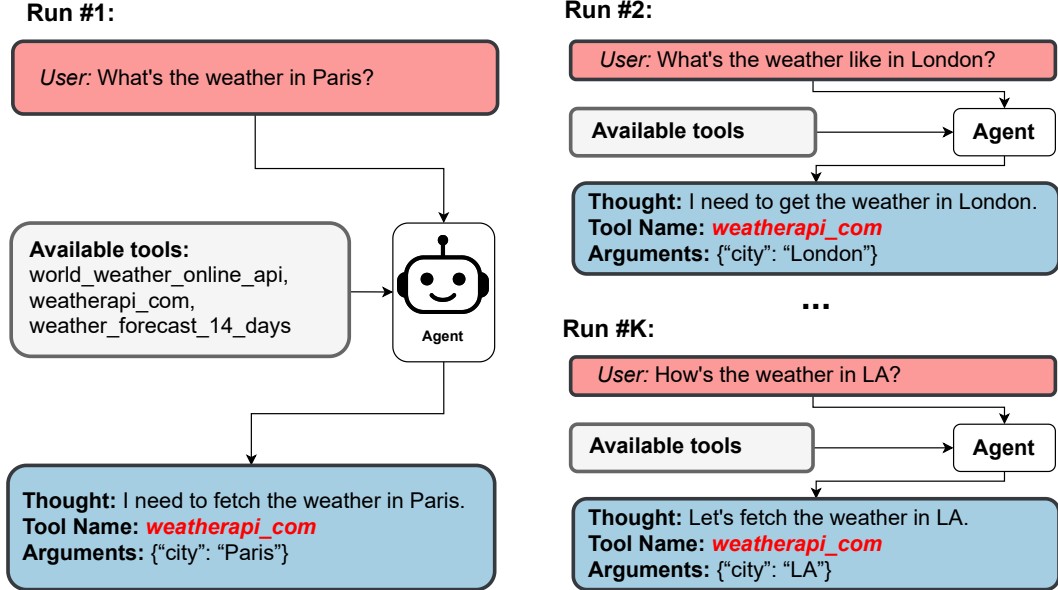

Figure 1: Tool-calling enables LLMs to act through external services, but the selection process introduces bias. Models may favor certain tools based on superficial metadata or position rather than relevance (here "weatherapi_com" is preferred), leading to a potential degraded user experience and unfair concentration of calls. If such biases are systematic across frontier LLMs, they risk distorting entire tool marketplaces, disadvantaging functionally equivalent competitors.

ing competing providers offering functionally equivalent services (RapidAPI, 2025a; Economize, 2024). See Appendix L covering the consequences of tool-selection bias in more detail.

In this paper, we make three key contributions.

**First**, we present a large-scale benchmark for measuring tool selection bias, which we use to conduct the first empirical study of tool-selection bias in LLMs. We introduce total variation–based metrics to quantify selection imbalances, assemble a novel benchmark of clusters of tools with equivalent functionality, and conduct extensive experiments across multiple models and configurations. These tell us that bias exists to a certain extent for *all* models tested; models either fixate on a single provider or disproportionately prefer earlier-listed tools in context.

**Second**, after confirming the existence of bias, we investigate the root causes of these biases. We define a rich set of tool-level features and use regression and permutation-importance analyses to identify which factors predict selection. We further perturb metadata fields to test their influence and probe whether continued pre-training on biased data can induce persistent preferences. Our findings show that semantic alignment between query and tool description is the strongest interpretable driver of tool choice, metadata interventions on descriptions reliably steer choices, and biased pre-training amplifies preferences. However, none of these fully explain the behavior.

**Third**, we propose a lightweight mitigation strategy: filtering the supplied tools to a subset of relevant ones and sampling uniformly among them. This approach substantially reduces bias while still ensuring the user task can be solved.

By uncovering, explaining, and addressing tool-selection bias, our work illuminates a critical blind spot in tool-augmented LLM research. Our contributions go beyond diagnosis: we provide reproducible resources and a simple mitigation strategy that practitioners can adopt immediately. More broadly, we aim to set a foundation for fairer, more reliable tool-calling systems and a precedent to judge tool-calling LLM applications not only by their accuracy but also by the equity of their interactions with external ecosystems.

## 2 RELATED WORK

**Fairness & Bias in LLMs.** Bias in large language models emerges from their opaque training on vast web-scale data, often carrying forward and amplifying human prejudices and reasoning short-cuts (Gallegos et al., 2024). Much of the literature has focused on identifying and mitigating these undesirable behaviors (Schick et al., 2021; Gallegos et al., 2024; Bouchard, 2024). Unwanted behaviors in LLMs manifest as social bias, where stereotypes shape outputs and lead to unequal treatment of protected groups, or as cognitive bias, reflecting human-like shortcuts such as anchoring or confirmation effects (Lou & Sun, 2024; Itzhak et al., 2024; O'Leary, 2025). Measuring these phenomena requires domain-specific metrics. For example, to measure how likely a model is to associate a neutral attribute with one group over another, one can utilize the normalized log-probability bias score (Kurita et al., 2019). More recently, formal certification frameworks have been proposed to quantify counterfactual bias over distributions of prompts with statistical guarantees (Chaudhary et al., 2025). Mitigation strategies are diverse and can be deployed at multiple stages (e.g., at pre-/post-inference via augmentation of training data or rewriting of outputs or during training via fairness-aware objectives) (Gallegos et al., 2024). However, no single approach suffices for every scenario; effective debiasing requires interventions matched to the task's fairness goals and deployment context. We are not aware of prior work on bias in the tool-use paradigm.

**Positional Bias in LLMs**. Prior research has revealed significant positional selection biases in how LLMs answer multiple-choice questions. Pezeshkpour & Hruschka (2024) and Zheng et al. (2024) demonstrated that LLMs exhibit sensitivity to the ordering of answer options, with performance varying systematically based on option position. This was corroborated in the investigation of Wei et al. (2024), who also found token sensitivity, where specific token characteristics at different positions affect selection. Recently Wang et al. (2025), analyses the underlying model mechanisms that give rise to position bias and proposed a mechanistic approach to eliminating it. In this work, we study and investigate a broad range of potential biases in the critical context of tool selection.

**Fairness in Tool Selection.** Research on tool-selection bias is limited; however, some works in information retrieval address related issues such as positional or ranking bias (Dai et al., 2024; Ziems et al., 2024). Similarly, in code generation, favoritism for certain providers has been observed (Zhang et al., 2025b). However, these studies do not examine the specific setting of tool selection among providers. Recent studies do show that LLMs can develop systematic preferences in this setting, for example, for selecting tools with manipulated metadata (Mo et al., 2025; Faghih et al., 2025; Zhang et al., 2025a). Analogously, one study shows an advanced attack on tool selection which splits malicious tool entries into fragments optimized for retrieval and selection, achieving high success rates while evading defenses (Shi et al., 2025). Other recent work has focused on adversarial robustness of tool selection, proposing statistical certification of tool-selection accuracy under adaptive attacks (Yeon et al., 2025). These findings reveal that tool-selection mechanisms are highly fragile and susceptible to both naïve and advanced exploitation. In contrast to overt adversarial attacks, our study focuses on subtler sources of bias: small, non-adversarial differences in phrasing or metadata that can still distort fairness in tool selection.

## 3 BIASBUSTERS: HOW WE UNCOVER, EXPLAIN, AND MITIGATE BIAS.

In this section, we present our end-to-end framework for uncovering and explaining tool-selection bias in LLMs. We start in Section 3.1 by introducing our formal definition of this bias. In Section 3.2, we describe how we generate a comprehensive benchmark of API clusters and user queries that enables systematic measurement of selection behavior. Section 3.3 details our analysis pipeline for explaining bias.

### 3.1 BIAS DEFINITION

Tool-selection bias captures the extent to which an LLM systematically favors certain APIs over others for reasons unrelated to their true utility. Formally:

**Definition 3.1** *Tool-selection bias is the systematic tendency of a model to favor certain APIs over others for reasons unrelated to the APIs' true relevance or utility for the task.*

To quantify this bias, we compare a model's empirical selection rates against an ideal uniform choice. Consider a cluster of $K$ APIs that can all solve a given query $q$. A perfectly unbiased model would select each API with probability $1/K$, forming a uniform distribution $U$. Let $P^{\mathrm{API}} = (P_1^{\mathrm{API}}, \ldots, P_K^{\mathrm{API}})$ denote the expected empirical selection-rate distribution over the cluster under all orderings of the API list. We measure cluster-level bias via the total variation distance

$$\delta_{\mathrm{API}} \;=\; \tfrac{1}{2} \sum_{i=1}^{K} \left| P_i^{\mathrm{API}} - \tfrac{1}{K} \right| \;=\; \mathrm{TV}\big(P^{\mathrm{API}},\, U\big).$$

Prior work has shown that list order itself can introduce bias: when multiple identical tools appear in sequence, the first position is favored Faghih et al. (2025). To capture this effect, we also record the expected selection-rate distribution over absolute positions, $P^{\mathrm{pos}}$, and compute $\delta_{\mathrm{pos}} = \tfrac{1}{2} \sum_{i=1}^{K} \left| P_i^{\mathrm{pos}} - \tfrac{1}{K} \right|$. Combining both yields an overall bias metric: $\delta_{\mathrm{model}} = \frac{\delta_{\mathrm{API}} + \delta_{\mathrm{pos}}}{2}$. Note that a high positional bias $\delta_{\mathrm{pos}}$ can be mitigated by randomizing API order at prompt time, whereas API-level bias $\delta_{\mathrm{API}}$ requires deeper intervention.

## 3.2 DATASET GENERATION

We build on the ToolLLM pipeline (Qin et al., 2024), which provides a large repository of APIs scraped from RapidAPI (RapidAPI, 2025b). To measure tool-selection bias, we construct a benchmark of functionally interchangeable APIs. Specifically, we cluster APIs into groups performing the same task (e.g., weather forecasting or translation), and generate balanced, provider-agnostic user queries that all APIs in a cluster can answer. The final benchmark consists of 10 clusters, each containing 5 APIs and 100 queries, yielding 1,000 total cluster-query pairs. Running LLMs on this benchmark produces empirical API-selection distributions, from which we compute our bias metrics $\delta_{\mathrm{API}}$, $\delta_{\mathrm{pos}}$, and $\delta_{\mathrm{model}}$. See Appendix A for clustering and query-generation details.

## 3.3 EXPLAINING BIAS

To pinpoint what drives tool-selection bias, we pursue three complementary analyses.

**Attribute-Level Analysis.** To test whether intrinsic API characteristics explain model preferences, we extract seven descriptive features for each API. Examples include semantic similarity between queries and descriptions, number of parameters, description length, readability, and promotional wording (See Table 3 for the full list). Our benchmark contains 10 clusters with 5 APIs each, giving 50 APIs in total. For each API, we measure its empirical selection rate and pair this with its feature values. This produces a dataset of 50 (API, features, selection rate) entries per model. We then probe relationships between features and selection behavior in three ways. First, we compute *Pearson correlations* between tool features and selection rate to capture linear and monotonic associations. Second, we fit a *linear regression* per model to quantify the aggregate explanatory power (reported as $R^2$) and inspect coefficients to understand the direction and relative weight of each feature. Third, we train *random-forest regressors* with cross-validation to allow for non-linear interactions and obtain alternative measures of feature importance. This pipeline reveals which API attributes most strongly influence the model's choices.

**Metadata Perturbation Experiments.** To isolate the superficial cues that drive tool-selection preferences, we apply a series of controlled perturbations to tool metadata. Specifically, we utilize the following manipulations: (1) *Full name scramble.* Replace every tool's name with a fresh 20-character random string, destroying any learned association tied to the literal name; (2) *Name shuffle.* Permute the tool names among APIs within each cluster so that names no longer align with their original endpoints; (3) *Single-tool perturbation.* Identify the most frequently chosen tool in each cluster and replace only its name with a random string; (4) *Description and parameter scramble.* Randomize each tool's descriptive text and parameter description(s) (but keep the original names intact) to test whether the semantic content beyond the name influences selection; (5) *Description-only / Parameter-only scramble.* Randomize only the tool descriptions (keeping parameter descriptions intact), or only the parameter descriptions (keeping the tool description intact), to disambiguate their individual contributions; (6) *Targeted description scramble (most-selected).* Identify the most frequently chosen API in each cluster and scramble only *its* description to test whether degrading its semantics reduces its selection share; (7) *Description transfer (most → least).*

```
Available tools:          Available tools:          Available tools:
  * API A                   * API B                   * API C
  * API B                   * API C                   * API A
  * API C                   * API A                   * API B

User:                     User:                     User:
What is the weather in Paris?  What is the weather in Paris?  What is the weather in Paris?
```

Figure 2: Cyclic rotations of one fixed tool list; each API appears at the top once.

Swap the most-selected API's description with the least-selected API while leaving other metadata untouched, assessing whether swapping semantic "advantage" transfers selection probability; and (8) *Full scramble.* Randomize each tool's descriptive text, parameter description(s), and tool names to test the effect of having minimal semantic signal in API metadata on bias.

By re-running our selection experiments under these alterations, we quantify how much of the observed bias is attributable to literal names, to deeper semantic content in descriptions and parameters, and to relative contrast between a clean and corrupted endpoint.

**Biased Continued Pre-Training.** We also test whether pre-training data itself can induce tool-selection bias. To verify this hypothesis, we perform biased continued pre-training (CPT) on Qwen3-8B using ∼3.5M tokens deliberately saturated with a single API's metadata. See Appendix C for additional details on corpus construction and training setup.

## 4 EXPERIMENTS

This section reports the empirical results from our experiments. We first describe the experimental setup (Section 4.1), then characterize tool-selection behavior and quantify bias using our metrics (Section 4.2). Next, we investigate the drivers of bias (Section 4.3), and finally, we evaluate an approach to mitigate any observed bias in Section 4.4.

### 4.1 EXPERIMENTAL SET-UP

**Models.** We evaluate a diverse set of LLMs, including GPT-3.5-turbo and GPT-4.1-mini by OpenAI (2025), Claude 3.5 Sonnet by Anthropic (2025), DeepSeek-V3.2-Exp (DeepSeek-AI et al., 2025), Gemini 2.5 Flash (Google, 2023), ToolLLaMA-2-7B (Qin et al., 2024), and Qwen3 models spanning 1.7B to 235B parameters (Bai et al., 2023).

**Reasoning.** Each model is prompted to produce a short chain of thought (Wei et al., 2022) followed by at most one tool call. This design makes outputs efficient to generate and easy to analyze.

**Parameter Setup & Dataset.** Experiments use our benchmark of 10 clusters, each containing 5 interchangeable APIs and 100 distinct user queries (Section 3.2 and Appendix A). Unless otherwise noted, decoding uses a temperature of $0.5$ and a top-p of $1.0$.

**API ordering.** As noted in previous work (Faghih et al. (2025)), LLMs prefer tools that appear earlier in the prompt. To control for this, we execute each query five times, each with a different cyclic rotation of a fixed API ordering (see Figure 2). This ensures that every API appears at the top for a given query exactly once.

### 4.2 HOW DO LLMS SELECT AMONG FUNCTIONALLY EQUIVALENT APIS?

Figure 3 shows our first results: the empirical selection distributions of six LLMs over three clusters of functionally equivalent APIs (see Appendix G.2 for the full figure including all clusters). Choices are far from uniform. In some clusters (e.g., geocoding), all models concentrate heavily on a single API, whereas in others (e.g., language identification) the distributions are flatter. We also observe that different models do not always prefer the same API, a point we explore further when analyzing alignment in selection behavior.

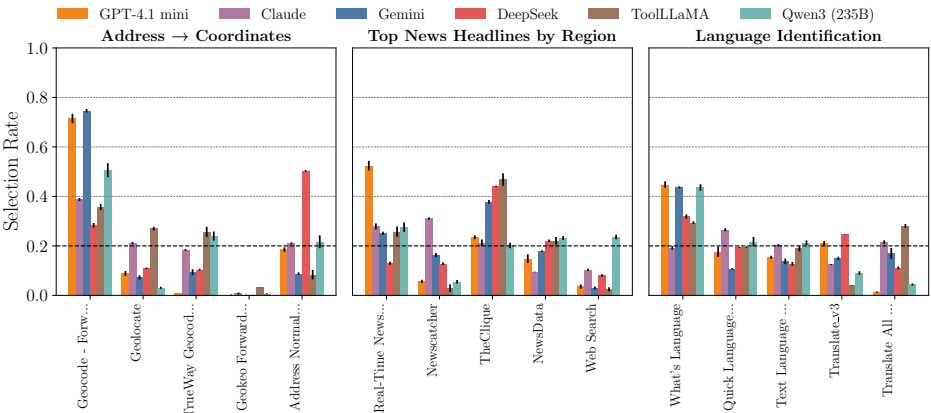

Figure 3: Selection distributions for six LLMs across three clusters of functionally equivalent APIs. Each subplot corresponds to one cluster, with the x-axis indicating the API in the cluster and the y-axis showing the (mean) fraction of times each model chose that API over 500 inference runs; error bars indicate the standard deviation across three independent experimental runs. The optimal uniform selection rate is highlighted.

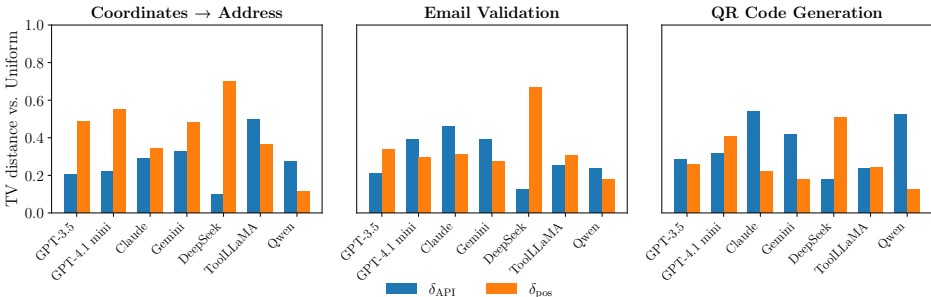

Figure 4: API- vs. positional bias by model for three clusters. Bars show total-variation deviation from uniform, where higher values indicate stronger bias.

**Positional bias persists even when API-level preference is weak.** Figure 4 summarizes API- vs. positional-bias (measured as total variation distance from uniform) across models and clusters. We notice two regimes: high API bias with low positional bias, or low API bias with high positional bias; a few exhibit both elevated. This pattern indicates that when no API clearly dominates, models rely more on positional cues.

**The extent of LLM bias in tool selection.** To get an idea of *how* biased each model is in their tool-selection behavior, we compute the positional bias $\delta_{\text{pos}}$, API bias $\delta_{\text{API}}$, and their average $\delta_{\text{model}}$ for seven different models (Section 3.1). Table 1 reports these metrics averaged across all clusters and three runs. All models exhibit substantial bias: $\delta_{\text{model}}$ values are around 0.3–0.4, meaning that roughly 30–40% of the selection probability mass would have to be redistributed to achieve fairness. GPT-4.1 mini is especially biased, with a combined metric of $\sim$0.38. The least biased model in our suite is Qwen 3 (235B), which attains the lowest $\delta_{\text{model}}$.

**LLMs are (mostly) aligned in their bias.** To examine the similarity of selection behavior of the different models, we represent each model by a vector obtained by concatenating its empirical selection distributions over APIs across all clusters, then compute pairwise Pearson correlations between these vectors; the resulting matrix is shown in Figure 15. It shows that many of the models share similar bias patterns: GPT 4.1-mini, Claude, Gemini, DeepSeek, and Qwen3 (235B) tend to favor and disfavor the same APIs. GPT-3.5 and ToolLLaMA stand apart with consistently lower cor-

Table 1: Average cluster-level API bias $\delta_{\mathrm{API}}$, positional bias $\delta_{\mathrm{pos}}$, and combined bias $\delta_{\mathrm{model}}$ (mean across clusters and runs $\pm$ std across runs).

| Model | $\delta_{\mathrm{API}}$ | $\delta_{\mathrm{pos}}$ | $\delta_{\mathrm{model}}$ |
|---|---|---|---|
| Gemini 2.5 Flash | $0.365_{\pm.003}$ | $0.306_{\pm.005}$ | $0.335_{\pm.002}$ |
| Claude 3.5 Sonnet | $0.370_{\pm.005}$ | $0.325_{\pm.005}$ | $0.347_{\pm.001}$ |
| DeepSeek-V3.2-Exp | $0.249_{\pm.003}$ | $0.504_{\pm.003}$ | $0.377_{\pm.001}$ |
| Qwen3 235B | $0.330_{\pm.006}$ | $0.168_{\pm.004}$ | $0.249_{\pm.001}$ |
| ToolLLaMA | $0.277_{\pm.002}$ | $0.391_{\pm.002}$ | $0.334_{\pm.002}$ |
| GPT-3.5-turbo | $0.320_{\pm.022}$ | $0.336_{\pm.012}$ | $0.328_{\pm.005}$ |
| GPT-4.1 mini | $0.331_{\pm.008}$ | $0.423_{\pm.002}$ | $0.377_{\pm.004}$ |

relations, suggesting qualitatively different selection behavior. This clustering of high correlations points to common drivers of bias. This could be due to a shared set of similar implicit decision rules. The relative divergence of GPT-3.5 and ToolLLaMA highlights that model architecture, capacity, or training objective can alter these tendencies, but overall alignment underscores that tool-selection bias patterns are not isolated quirks but mostly reproducible phenomena across LLMs.

**Ablation and Sensitivity Analysis.** We also analyze the robustness of tool-selection bias across several factors (full results in Appendix E). *Temperature:* raising temperature reduces bias modestly by softening extreme preferences. *Top-p:* increasing top-p has only a negligible effect. *Model size:* larger models exhibit noticeably less bias than smaller ones. *API ordering:* cyclic vs. random permutations produce very similar outcomes, indicating intrinsic preferences dominate. *System prompts:* rewording or restructuring prompts shifts which tools are favored, but does not remove bias. *Toolset size:* the composition of the toolset given in context is the primary driver of selection behavior, more than the size of the toolset alone (see Appendix J).

### 4.3 ARE TOOL-SELECTIONS DRIVEN BY HUMAN-INTERPRETABLE HEURISTICS?

We investigate the drivers of bias along three complementary axes: (i) feature-level correlations between API attributes and selection rates, (ii) direct interventions on API metadata, and (iii) biased continued pre-training (CPT) to test whether exposure alone can plant preferences. We summarise our results below, with full details in Appendix F.

**Which API-level features predict selection rates?** The results show a consistent pattern: (1) Semantic similarity between queries and API / tool descriptions is the strongest predictor of selection (Table 4). By contrast, structural or stylistic attributes (e.g., parameter count, promotional wording) exhibit little consistent influence. (2) Linear regression reveals that surface-level semantic alignment is the primary signal but leaves a lot unexplained ($R^2 < 0.4$ as can be seen in Figure 13), and (3) Random forests fail to offer meaningful improvement.

**How do metadata interventions affect API selection?** Figure 5 shows the TV distance from the base selection distribution to the distribution obtained after a certain metadata perturbation, averaged over clusters and experimental runs. It shows a clear hierarchy: logically minimizing semantic signal by scrambling the description, name, and parameters causes the biggest shift. For Gemini, corrupting descriptions+parameters yields the second largest, most reliable shifts in selection (0.450 ± 0.203 TV), followed by scrambling the description of the most popular tool (0.419 ± 0.179), and even description swaps meaningfully steer choices (0.338 ± 0.205). By contrast, name-only perturbations are smaller and have higher variance, and parameters-only are the least impactful. We see similar results for GPT; description perturbation is more impactful than name/parameter perturbations. Overall sensitivity is higher for Gemini than GPT (mean TV 0.310 vs 0.234), indicating greater responsiveness to metadata changes.

We show similar results across clusters in Figure 14; however, we observe that manipulating the description of certain tools has mixed effects from cluster to cluster. It tells us that the impact of

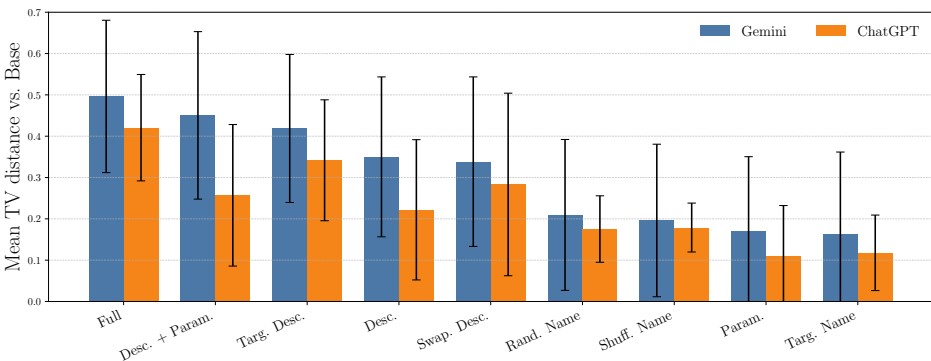

Figure 5: Mean total-variation (TV) distance from the base selection distribution (no perturbation) to the distribution pertaining to each metadata perturbation (higher = larger shift). Blue bars show results for Gemini and orange bars for GPT-3.5. The error bars denote standard deviation across clusters. The run-to-run standard deviation is left out; max run-to-run variability of per-run mean TV was $0.084$.

description tampering is context-dependent: the same intervention can invert, redistribute, or barely change preferences, underscoring that tool choice emerges from multiple interacting cues.

Together, these patterns indicate that description-level semantics are the primary cues models use to discriminate among functionally similar APIs. Name perturbation alone tends to inject noise without consistent effects. Finally, bias persists under minimal semantic signal (only leaving certain parameter schema fields intact), implying selection behavior sometimes relies on residual, non-obvious priors rather than solely on coherent, human-interpretable heuristics. For an analysis of the impact of metadata perturbations on bias, see Appendix H.

**Does additional pre-training exposure to one endpoint change selection distribution?** We run biased continued pre-training on Qwen3-8B in which the training corpus is saturated with metadata for a single target endpoint (Text Language by API-Ninjas)[1]. All inference settings, prompts, and decoding parameters are held fixed; only the checkpoint changes (see full training and inference hyperparameters in Appendix I). We then compare selection rates within the same Language Identification cluster before CPT and after $\sim$1/3, $\sim$2/3, and one full epoch of biased training (See Figure 17 for full figure). We observe that biased CPT substantially increases the selection of the exposed endpoint but does not fully determine choice. The target endpoint's share rises from 0.006 (base) to 0.122 after 1/3 epoch, remains 0.122 at 2/3 epoch, and nudges to 0.128 at one epoch. This is an absolute gain of $\sim$12 percentage points (over $20\times$ relative). Most of the shift is realized early, suggesting a quickly saturating response. This demonstrates that biased exposure during training can directly shape tool-selection preferences in favor of the exposed endpoint. However, since the target never approaches a dominant share, pre-training exposure explains only part of the bias, and additional factors still shape tool choice.

### 4.4 CAN WE MITIGATE THE OBSERVED BIAS?

After demonstrating the existence and causes of bias, we seek to mitigate it. Our approach is simple: models often recognize which APIs can solve a certain task, but still exhibit skewed preferences among interchangeable endpoints. Hence, we propose to decouple recognition from selection via a lightweight debiasing module. This module uses a smaller LLM (Qwen3-14B), prompted to return only the subset of APIs from the candidate list that can solve the given query. We then choose uniformly at random from this subset, ensuring that each valid API has the same expected probability of selection. This eliminates positional or metadata-based favoritism while maintaining task coverage. See Appendix D for benchmark details and evaluation metrics. This section showcases the potential of the method and outlines how it can help.

---

[1]We use this model because its smaller size makes training tractable on modest hardware.

Table 2: Subset–selection performance for Qwen3 (14B) (overall and by ground–truth set size $K$).

| Overall | | By ground–truth set size $K$ | | | | |
|---|---|---|---|---|---|---|
| | | $K$ | n | Precision | Recall | Exact Set Match |
| Micro-Precision | 0.9964 | 2 | 300 | 1.0000 | 0.7717 | 0.5433 |
| Micro-Recall | 0.8856 | 3 | 200 | 0.9925 | 0.8850 | 0.7350 |
| Exact Set Match Rate | 0.6900 | 4 | 300 | 0.9940 | 0.9633 | 0.9100 |
| | | 5 | 200 | 1.0000 | 0.8610 | 0.5350 |

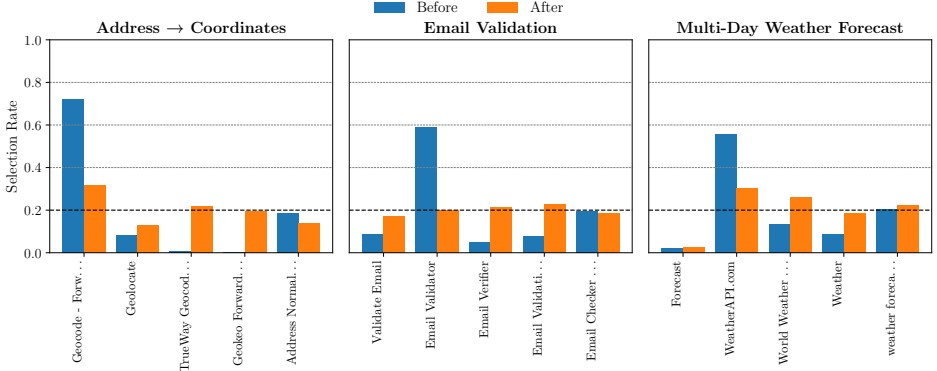

Figure 6: Selection distributions for GPT-4.1 mini with and without utilizing our mitigation method across three clusters. Each subplot corresponds to one cluster, with the x-axis indicating the API in the cluster and the y-axis showing the fraction of times that API was chosen using the corresponding setup over 500 inference runs.

**Subset selection avoids spurious inclusions of incorrect APIs while maintaining high coverage of correct APIs, thereby mitigating bias without sacrificing performance.** Table 2 summarizes results for Qwen3-14B as the subset selector. Overall micro-precision is ∼1.00 (0.9964), meaning the selector almost never adds tools to the candidate set that cannot solve the user's task. This is a desirable property as it means that our mitigation module is unlikely to output an incorrect tool, and therefore, performance will not suffer. Note that since all ground-truth set size classes have nearly the same number of queries, micro- and macro-precision are effectively equivalent; we report micro-precision for simplicity. Micro-recall is ∼0.89 (0.8856), so on average the selector itself is not biased and retains most ground-truth tools, with an exact set match of 0.69 across all instances. Broken out by the size of the ground-truth set $K$, precision remains essentially perfect across the board, while recall varies: it is strongest at $K$=4 (0.9633) and somewhat lower at $K$=2 (0.7717) and $K$=5 (0.8610). The corresponding exact-match rates reflect the same pattern (notably 0.9100 at $K$=4). In practice, this means the subset filter very rarely introduces distracting tools (good for performance), but it can occasionally omit a true option when $K$ is small or large. Taken together, these results suggest subset selection is a promising first line of defense: it does not lead to spurious inclusions (high precision) and maintains high coverage of correct APIs on average, and thus, less bias downstream (strong recall).

**Our mitigation method successfully alleviates tool-selection bias.** Figure 6 shows the empirical selection distributions of GPT-4.1 mini before and after applying our mitigation method over three clusters of functionally equivalent APIs. Where choices were far from uniform before, after the mitigation method is applied we notice an even spread of selection share. However, in the weather forecasting cluster (right), we see that one API is still being underutilized, perhaps being an indication that the model has a difficult time using this API correctly even when it is the only API available. The effectiveness of the mitigation method is further exemplified in Table 5, where a steep decrease in all our bias metrics after applying the mitigation method can be observed.

## 5 CONCLUSION

In this paper, we introduced the first benchmark for evaluating tool selection bias in LLMs. Our results establish tool selection bias as a real and potentially significant issue for tool-augmented (agentic) LLMs, with implications for user experience, operational cost, and marketplace fairness. This study offers a concrete starting point for understanding and mitigating this bias.

**Limitations:** This study is limited by using 100 synthetically generated user queries per cluster and a narrow seven-feature set for selection-rate predictions, limited model/repeat coverage, and a focus on APIs from RapidAPI and English queries. Although our choices were constrained by compute (our setup already required ∼500,000 inference runs), these factors may induce variance and restrict generality.

**Future Work:** Future work should scale queries and clusters and enrich features with deeper semantic and structural signals to raise predictive power beyond the modest observed $R^2$. In addition, deploying more expressive models (e.g., boosted trees or deep nets) with cross-validation could capture higher-order interactions between tool features, further increasing their explanatory power.

We also believe it would be valuable to isolate the contribution of preference tuning (e.g., RLHF) to tool-selection bias, since this might implicitly reward particular writing styles or phrasings in tool names/descriptions and thereby skew selection among otherwise equivalent tools. Finally, richer analysis of models' internal decision processes could strengthen interpretability: while we now allow a single reasoning step before tool choice, preliminary inspection suggests models often do not explicitly justify why a specific tool is selected. Allowing longer reasoning traces and conducting more systematic chain-of-thought analysis may yield clearer causal explanations of selection behavior, but we leave this for future work due to computational constraints. Lastly, broader replication across LLMs and runs would aid in quantifying variability.

## ACKNOWLEDGMENTS

JY acknowledges support from Microsoft Ltd. AB and PT acknowledge the 2025 UK AISI Systemic Safety Grant and the UKRI Turing AI Fellowship (EP/W002981/1). AB, JY and PT are also affiliated with the Institute for Decentralized AI, which is supported by an AI Safety Fund grant. This work is supported by the European commissions' DVPS project. This work is supported by the Schmidt AI 2050.

ETHICS STATEMENT

This research did not involve identifiable human data or animals and therefore did not require approval from an institutional ethics committee or review board. All experiments are conducted for scientific purposes only. The work does not involve or target any sensitive attributes such as gender, race, nationality, or skin color. Our study focuses on identifying and minimizing tool-selection bias in LLM agents, with the aim of improving the trustworthiness and safety of the deployment of LLM agents.

REPRODUCIBILITY STATEMENT

We have made every effort to ensure the reproducibility of our work. We provide detailed descriptions of data and experimental setup in Section 4. Our code is available as a public GitHub repository to facilitate replication.

**Additional note.** During initial experiments we observed an unusually large standard deviation in DeepSeek's selection distributions for certain clusters. When we re-ran the experiments under the same configuration, this anomaly disappeared. Since we use the generic `deepseek-chat` endpoint for our LLM calls, which always serves the latest model version, one of the later runs was executed after a silent model update and thus, ran with a different model. All DeepSeek results reported in the current version of the paper are based on a fresh set of runs collected using the same model, which removes the anomalous error bars.

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

# A   MORE DETAILS ON DATA GENERATION

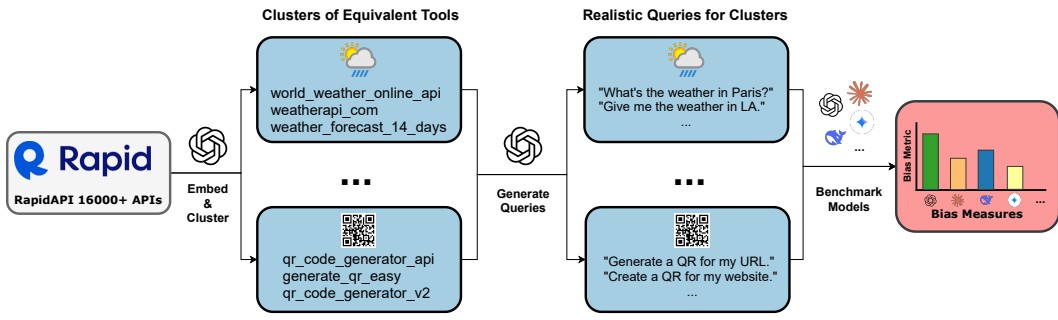

Figure 7: An overview of our clustering and query generation pipeline.

We build on the "tool-usage" evaluation pipeline introduced by Qin et al. (2024), hereafter referred to as ToolLLM, owing to its wide adoption and extensibility. At its core, ToolLLM assembles a large catalog of real-world APIs scraped from RapidAPI spanning 49 functional categories (RapidAPI, 2025b). For each API, ToolLLM provides a JSON file containing the API's human-readable name, detailed description, and full parameter schema. In this work, we leverage exactly that API repository but restrict our attention to the stages in which ToolLLM selects among a short list of retrieved candidates. Note that in ToolLLM, a set of closely-related APIs is called a 'tool'. For example, a geocoding tool could offer both forward geocoding and reverse geocoding APIs[2].

We assemble our benchmark in two stages: (1) clustering APIs into functionally equivalent groups, and (2) generating realistic user queries for each group (see Figure 7).

---

**Algorithm 1** Generation of Functionally-Equivalent API Clusters

---

**Require:** API id to metadata map $\pi$,
   Precomputed embeddings $E$,
   List of "general" APIs $G = \{(\text{tool}, \text{tool\_desc}, \text{api\_name}, \text{api\_desc})\}$,
   neighbor count $K$,
   max outlier loops $R$
1: $\mathcal{C} \leftarrow \emptyset$
2: **for all** (tool, tool\_desc, api\_name, api\_desc) $\in G$ **do**
3:    Construct query text $q = $ " tool : tool\_desc | api\_name : api\_desc "
4:    Embed $q$ with ADA: $\mathbf{v}_q \leftarrow \text{Embed}(q)$
5:    Compute cosine similarities: $s_i \leftarrow \cos(\mathbf{v}_q, E_i) \quad \forall i$
6:    Select top–$K$ unique tools with largest similarity and store in set $\text{TOP}_K$
7:    candidate $\leftarrow \{\pi[i] \mid i \in \text{TOP}_K\}$
8:    **for** $r \leftarrow 1$ **to** $R$ **do**
9:       Prompt GPT-4 to detect outliers: outliers $\leftarrow \text{DetectOutliers}(\text{candidate})$
10:       **if** outliers $= \emptyset$ **then break**
11:       **end if**
12:       Remove outliers from candidate: candidate $= $ candidate $\setminus$ outliers
13:    **end for**
14:    **if** $|\text{candidate}| > 3$ **then**
15:       $\mathcal{C} \leftarrow \mathcal{C} \cup \{\text{candidate}\}$
16:    **end if**
17: **end for**
18: **return** $\mathcal{C}$

---

[2]Geocoding is the process of converting between human-readable addresses and geographic coordinates: "forward" geocoding maps an address (e.g., "1600 Amphitheatre Parkway") to its latitude/longitude, while "reverse" geocoding maps a given coordinate pair back to a structured postal address.

```
You are a prompt-writing assistant. I will give you a set of API
    endpoints (tool name + description, endpoint name + description,
    and potentially the required parameters) that all perform the same
     underlying task. Please generate exactly {n} distinct, natural-
    language user queries that could be satisfied by ALL of these
    endpoints. **Include realistic sample values** for any required
    parameters (e.g. use "https://example.com" for a URL, or "Hello
    World" for a text field). Return them as a JSON array of strings,
    with no extra commentary.

User:
Here are the endpoints:
- Tool: WeatherNow - Provides current weather information
  Endpoint: Current Weather - Returns temperature, humidity,
  and conditions for a given location.
  Required parameters:
    * city (string) - name of the city
    * country (string) - ISO country code
- ...
```

Figure 8: Example prompt used for query generation. The model outputs $n$ natural-language queries that all listed endpoints can satisfy.

**API clustering.** We begin by embedding every endpoint's metadata (tool name, API name, descriptions, etc.) into a shared vector space using a pre-trained text encoder (OpenAI's `text-embedding-ada-002` model). We then curate a small set of "seed" APIs whose descriptions span a number of "general" tasks, such as text translation or weather forecasting. For each seed, we retrieve its top-$K$ nearest neighbors in embedding space to form a candidate cluster. To ensure true functional equivalence, we iteratively prompt GPT-4 to flag any outlier endpoints that cannot perform the same task as the rest; flagged APIs are removed and the check repeats for up to a pre-defined number of rounds. Any cluster that stabilizes with more than three members is retained. See Algorithm 1 for an overview of our clustering approach. Lastly, we manually inspect and refine these clusters, yielding 10 high-quality groups of five APIs each.

**Query generation.** For each cluster, we prompt GPT-4 (see Figure 8) to produce natural-language queries that all members can satisfy. In batches of ten, the model generates candidate queries until we collect 100 unique queries per cluster, filtering out duplicates. In cases where freeform generation exhibits provider-specific bias (e.g., mentioning a particular vendor's feature), we switch to a template-filling workflow: we design a small set of generic templates with placeholders (e.g. "Get the latest news headlines for {country} about {topic}."), and ask GPT-4 to instantiate each template multiple times with realistic sample values.

**Final curation.** All 1,000 generated queries are then reviewed by hand to remove any that inadvertently favor a single provider or rely on specialized parameters. The resulting dataset consists of 10 clusters with 5 APIs each, and 100 balanced, provider-agnostic queries for each cluster.

Running each model over these prompts yields empirical selection distributions over APIs and list positions, from which we compute our total-variation-based bias metrics $\delta_{\text{API}}$, $\delta_{\text{pos}}$, and $\delta_{\text{model}}$. This rigorously grounded benchmark enables precise measurement and comparison of tool-selection bias across models and settings.

## B ATTRIBUTE-LEVEL ANALYSIS FEATURE TABLE

See Table 3 for the list of features used in the analysis of Section 4.3.

## C MORE DETAILS ON THE BIASED CPT EXPERIMENT

We test whether pre-training data can cause tool-selection bias by doing biased continued pre-training (CPT) on a single model. That is, we do additional next-token training on raw text using

Table 3: API-level predictor features.

| Feature | Description |
|---|---|
| avg_similarity_tool_desc | Mean text similarity between cluster queries and the tool's description. |
| avg_similarity_api_desc | Mean text similarity between cluster queries and each API's description. |
| age_days | Days since the API was first published. |
| desc_name_length_sum | Total character count of the API's name plus description. |
| num_params | Number of required and optional parameters. |
| flesch_reading_ease | Flesch reading-ease score of the combined descriptions. |
| positive_word_count | Count of positive or promotional words (e.g. "efficient," "robust"). |

~3.5M tokens deliberately saturated with one endpoint's metadata (its name, description, and parameter info). After this exposure, we re-run the selection tasks and measure shifts in that endpoint's selection share.

To generate the biased corpus, we synthesize long-form prose with an external LLM (Gemini 2.5 Flash). We prompt it to produce a single document of roughly 1.1–1.3k words written in a randomly sampled style (e.g., blog note, Q&A memo, release note, how-to guide, troubleshooting checklist). The prompt requires frequent natural mentions of the target API name, the exact or faithfully paraphrased tool description, and the exact or paraphrased API description; it also requests inclusion of the endpoint path in about 60% of documents and parameter metadata in about 50%. This pipeline yields a large, stylistically varied corpus that is nevertheless saturated with the same endpoint's metadata.

We then run CPT on the same base model used elsewhere (i.e., Qwen3-8B) with parameter-efficient adapters (LoRA) attached (see Hu et al. (2022) for more details on LoRA). We keep tokenizer unchanged.

We evaluate pre/post CPT selection distributions on the original cluster, prompts, and inference settings. The primary outcome is the shift in the target API's selection share. We also measure spillover: changes in the selection shares of non-target APIs within the cluster. If biased CPT reliably increases the target API's selection share, this is evidence that a portion of tool-selection bias originates from pre-training exposure.

## D  DETAILS ON THE IMPLEMENTATION AND EVALUATION OF THE MITIGATION METHOD

After showing the existence and possible causes of bias, we seek to mitigate it. We pursue a simple approach based on the following insight: Models often know which APIs can solve a task but can possibly exhibit biased choices among interchangeable endpoints. We decouple capability recognition from final selection via a lightweight debiasing module.

The debiasing module consists of a lightweight LLM (Qwen3 14B in our case) prompted to output the subset of APIs from the given candidate list that can solve the task given in the query. This way, we get a subset selector that outputs an array of the APIs that can complete the task. The system prompt constrains the output to an exact list with no prose. From the returned set $S$, we pick one API uniformly at random. This API then replaces the original API list and is used for the rest of the tool-usage pipeline.

If this approach is successful, each API in $S$ has an expected selection share of $1/|S|$, eliminating position/API favoritism at the choice stage. If the selector's true positive/negative rates are high enough, the overall selection distribution approaches uniform even when original models were skewed. This, following our definition, means the tool selection stage becomes unbiased by design.

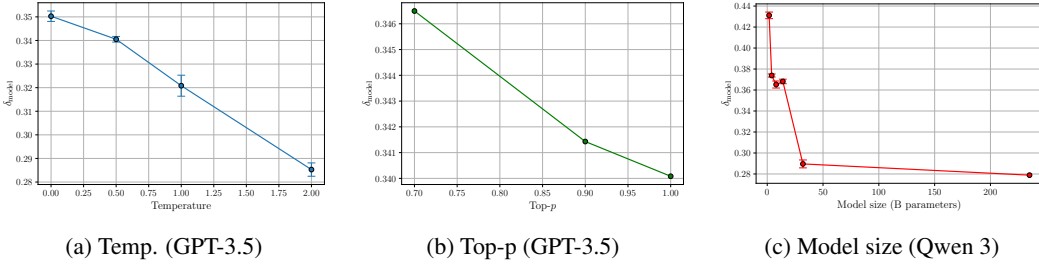

(a) Temp. (GPT-3.5)        (b) Top-p (GPT-3.5)        (c) Model size (Qwen 3)

Figure 9: Sensitivity of the combined bias metric $\delta_{\text{model}}$ to model hyperparameters. Each point is the mean over three independent runs (except for the top-p subplot); vertical bars show one standard deviation where available.

To evaluate this approach, we build a 1000-query benchmark with 8 API candidates per query and a ground truth set indicating which $K \in \{2, 3, 4, 5\}$ APIs are sufficient ($\sim$250 items each). We report subset quality: precision, recall, and exact-set match. Note that the formula for exact-set match is given by $\frac{1}{N} \sum_{i=1}^{N} \mathbf{1}[S_i = G_i]$ where $G$ denotes the set of ground truth sets, $S$ the set of selected subsets, and $N$ is the number of queries. Additionally note that bias can persist if the subset selector itself is biased; underselecting viable tools (false negatives) or selecting unrelevant ones (false positives). Measures of recall and precision will tell us whether this is the case.

## E   MORE ELABORATE ABLATION AND SENSITIVITY ANALYSIS

**Temperature.** Raising temperature reduces combined bias. As shown in Figure 9a, as temperature goes from 0 to 2, the mean $\delta_{\text{model}}$ for GPT-3.5 drops from about 0.350 to 0.285, a 6.5% absolute reduction. Figure 10 makes clear why: the overall selection patterns remain similar across temperatures, but higher temperatures soften extreme preferences. This suggests that increased stochasticity slightly mitigates bias, but does not eliminate it.

**Top-p.** Figure 9b shows how the combined bias $\delta_{\text{model}}$ for GPT-3.5 varies with the top-p cutoff. Increasing top-p from 0.7 to 1.0 yields a small decrease in bias (from $\sim$0.346 to $\sim$0.340), suggesting that less aggressive truncation of the probability distribution slightly softens extreme tool preferences. The effect is noticeably weaker than the temperature change.

**Model Size.** In Figure 9c, the combined bias $\delta_{\text{model}}$ is depicted for Qwen 3 with varying model size. It seems that larger models exhibit less bias, with a notable drop at 32B. This pattern suggests that larger models develop more nuanced selection mechanisms which temper extreme preferences for certain APIs.

**API Ordering.** Figure 11 compares GPT-3.5's API selection under two different ordering schemes: cyclic rotations versus random permutations. Across all clusters, the choice distribution is very similar: no API's selection rate shifts more than about ten percentage points. This indicates that the ordering of the APIs has some influence, but the dominant signal is the model's intrinsic preference. However, the small differences could also reflect the inherent noise from stochastic token sampling, and overall we argue that the tool-selection behavior is robust to either type of shuffling.

**System Prompts.** To evaluate how sensitive tool selection is to the phrasing and structure of the instructions given, we compare three variants of the system prompt: the original "Base" prompt, a lightly reworded "Similar" prompt, and a structurally different "Adjusted" prompt. Figure 12 shows the resulting distributions for GPT-3.5.

Prompt wording shifts model preferences but does not remove bias. Reworded prompts can amplify dominant choices and in some cases radically redistribute the selection shares. Elsewhere, effects are modest. Overall, framing and formatting can tilt the implicit ranking among functionally equivalent APIs, indicating that part of the observed bias is prompt-dependent even as a tendency to favor a subset of tools remains.

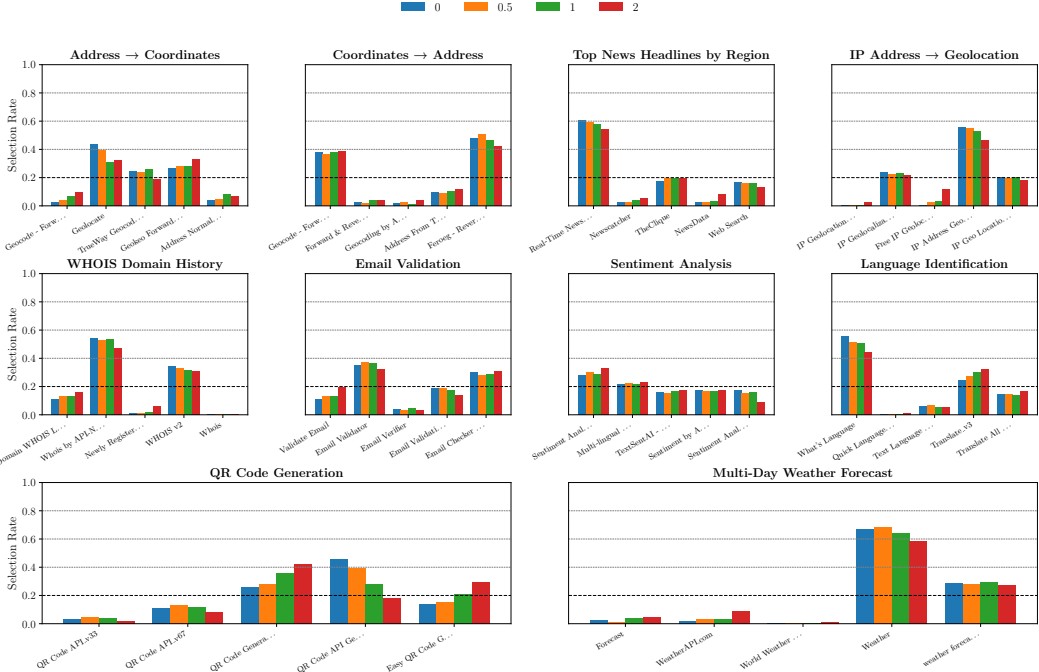

Figure 10: Selection distributions for GPT-3.5 using four different temperatures across ten clusters of functionally equivalent APIs. Each subplot corresponds to one cluster, with the x-axis indicating the API in the cluster and the y-axis showing the fraction of times that API was chosen by the respective model over 500 runs.

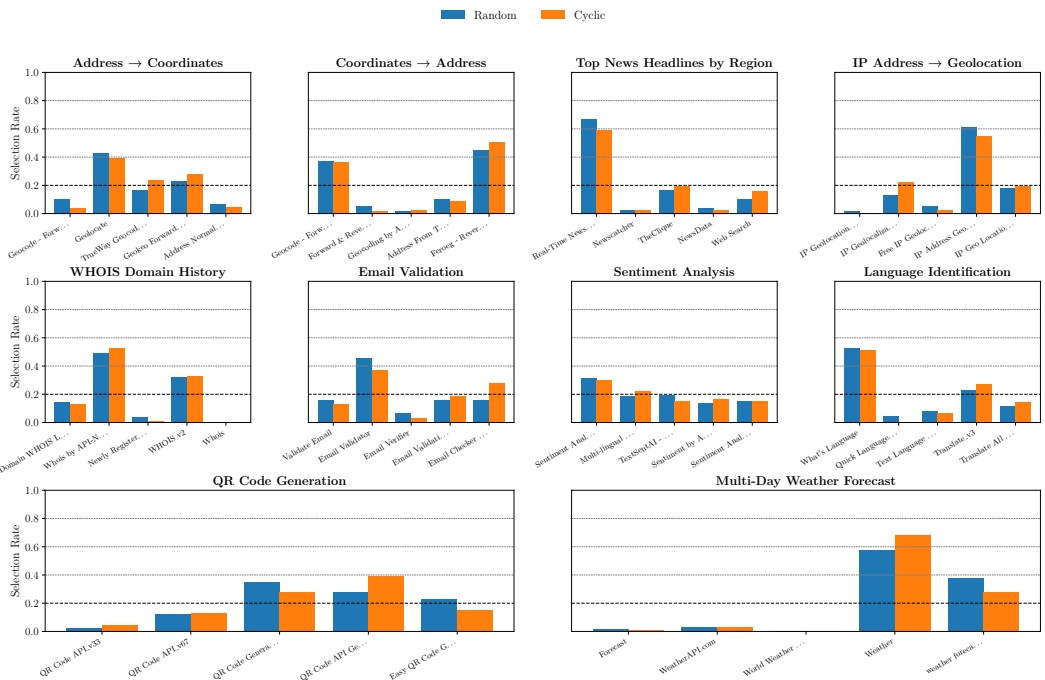

Figure 11: Selection distributions for GPT-3.5 using cyclic and random shuffling of the APIs across ten representative clusters. Each subplot corresponds to one cluster, with the x-axis indicating the API in the cluster and the y-axis showing the fraction of times that API was chosen for that specific API ordering method over 500 runs.

## F    More Elaborate Discussion on the Explanation of Bias

We now expand on the analysis given in the main text surrounding the investigation of bias. We expand on the feature-level analysis, where we try to predict selection rates according to intrinsic API attributes, and on the perturbation experiments that directly intervene on the API metadata to see which cues the models rely on during selection.

### F.1    Which API-level features predict selection rates?

We extract a common set of descriptive features from every API (see Section 3.3) and mean-center them to investigate how being relatively high or low on a feature affects the API selection. These are then paired with the empirical selection rates yielding a dataset of 50 examples for each LLM. We then probe relationships between features and selection behavior in three ways. First, we compute Pearson correlations to capture linear and monotonic associations. Second, we fit a linear regression per model to quantify the aggregate explanatory power (reported as $R^2$) and inspect coefficients to understand the direction and relative weight of each feature. Third, we train random-forest regressors with cross-validation to allow for non-linear interactions and obtain alternative measures of feature importance.

**Similarity between tool / API description and query is most correlated to selection rate.** As Table 4 makes clear, the most predictive feature of API selection is semantic similarity between the query and the tool/API descriptions. Both avg_similarity_tool_desc and avg_similarity_api_desc are consistently positively correlated with selection rates—especially strong for Qwen and clear for Gemini; GPT shows the same pattern, albeit weaker with higher p-values. By contrast, structural or stylistic attributes (e.g., parameter count, promotional wording) exhibit little consistent influence. Tool age (age_days) shows a modest, broadly consistent negative correlation across models.

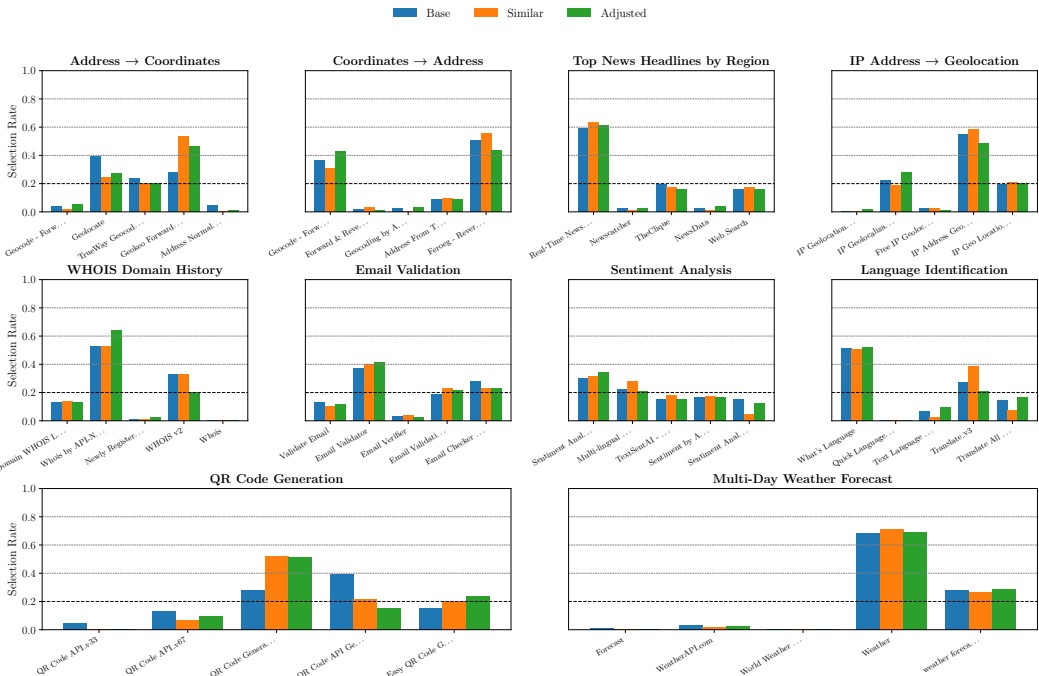

Figure 12: Selection distributions for GPT-3.5 using different system prompts across ten clusters. Each subplot corresponds to one cluster, with the x-axis indicating the API in the cluster and the y-axis showing the fraction of times that API was chosen using the corresponding system prompt over 500 runs.

Table 4: Correlation between API-level features and model selection rates. Each entry shows Pearson $r$ with its $p$-value.

| Feature | GPT-4.1 mini | Gemini | Qwen |
|---|---|---|---|
| avg_similarity_tool_desc | $+0.227$($p$=0.113) | $-0.092$($p$=0.526) | $+0.234$($p$=0.101) |
| avg_similarity_api_desc | $+0.111$($p$=0.442) | $+0.330$($p$=0.019) | $+0.411$($p$=0.003) |
| age_days | $-0.199$($p$=0.201) | $-0.144$($p$=0.356) | $-0.163$($p$=0.296) |
| desc_name_length_sum | $+0.044$($p$=0.760) | $+0.103$($p$=0.477) | $+0.038$($p$=0.795) |
| num_params | $-0.065$($p$=0.653) | $+0.038$($p$=0.793) | $-0.185$($p$=0.198) |
| flesch_reading_ease | $+0.160$($p$=0.267) | $+0.176$($p$=0.222) | $+0.098$($p$=0.496) |
| positive_word_count | $+0.126$($p$=0.384) | $+0.087$($p$=0.547) | $+0.093$($p$=0.521) |

**Linear regression reveals that surface-level semantic alignment is the primary signal but leaves a lot unexplained.** Figure 13 tells us that linear models explain only part of the variance: $R^2$ is modest—0.143 for GPT-4.1 mini and 0.387 for ToolLLaMA—leaving substantial error. Coefficients show surface-level semantic alignment dominates: similarity between the query and tool/API descriptions has the largest positive weights for most models, with Qwen weighting both most strongly and Gemini emphasizing API-level descriptions. Unexpectedly, ToolLLaMA gives a negative weight to tool-description similarity. Other features contribute little. Hence, semantic alignment is important in driving selection but still gives an incomplete explanation, implying nonlinear or omitted factors and motivating more flexible models (e.g., random forests).

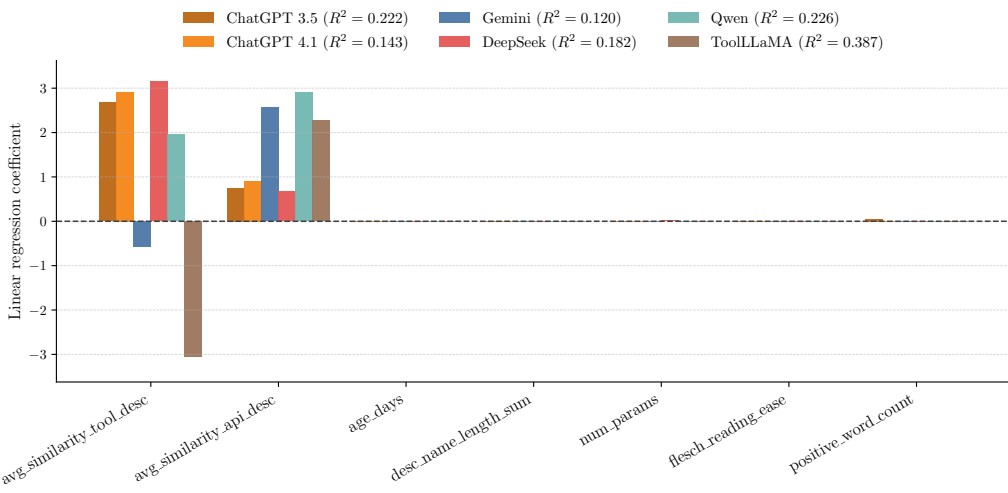

Figure 13: Linear regression feature weights used to predict API selection rates for six LLMs. Each group of bars corresponds to one API-level feature; different colors denote models, with their $R^2$ shown in the legend. Larger positive weights indicate features that increase the predicted selection rate.

**Random forests fail to offer meaningful improvement.** We fitted random-forest regressors with the same mean-centered features using both cross-validation and a held-out split, but predictive performance was poor, meaning the forests often did worse than a trivial constant baseline. This suggests the available features, at least in their current form and scale, don't contain enough signal or that some artifacts are overwhelming the gains from nonlinearity. Therefore, any feature-importance estimates from these trees would be unreliable and we do not lean on them for explanation. Future work could involve revisiting this with a richer feature set, more data, or alternative nonlinear modeling.

### F.2 How do metadata interventions affect API selection?

We saw that corrupting descriptions produces much larger and more stable effects on selection behavior than name-level perturbations, which are sometimes noisy and unpredictable, in Section 4.3. Figure 14 corroborates this across clusters. Name perturbations often leave distributions near-unchanged or can make them drift unpredictably (e.g., Cluster 1, where Cluster 1 is positioned at the top-left and Cluster 6 at the bottom-right), whereas description/parameter scrambles frequently overhaul rankings: sometimes amplifying the dominant API (Cluster 2), other times causing dramatic re-ordering (Clusters 3). Name edits rarely produce comparably stable re-ranking.

Together, these patterns indicate that description-level semantics (and, to a lesser extent, parameter semantics) are the primary cues models use to discriminate among functionally similar APIs. Name perturbation alone tend to inject noise without consistent effects. Finally, bias persists under minimal semantic signal (only names and schema fields), implying selection behavior sometimes relies on residual, non-obvious priors rather than solely on coherent, human-interpretable heuristics.

**Manipulating the description of certain tools has mixed effects across clusters.** Figure 14 (lower row) shows three behaviors when we manipulate descriptions. First, swapping the most- and least-popular tools' descriptions can invert their selection rates (Cluster 4), indicating description text alone can dominate choice. Second, the same swap sometimes yields only a modest lift for the least-popular tool while unexpectedly altering the selection shares of unaffected tools (Cluster 5), suggesting the landscape is reconfigured rather than ranks simply exchanged. Third, in some clusters the swap has minimal effect (Cluster 6), implying other cues—e.g., name priors or parameter schemas—anchor preferences.

Targeted corruption of the most-selected tool's description has similarly inconsistent effects. In Cluster 4, scrambling collapses its share to near zero as another tool absorbs the mass; in Clusters 5–6, corruption diffuses probability across competitors, producing a more even allocation. Overall, description tampering often wields substantial influence, but the impact is context-dependent: the same intervention can invert, redistribute, or barely change preferences, underscoring that tool choice emerges from multiple interacting cues.

## G Additional Figures

### G.1 Correlation in Selection between Models

Figure 15 shows that models exhibit varying degrees of correlation in their tool-selection patterns, suggesting shared but non-identical biases across families.

### G.2 Selection Distributions for all Clusters

This subsection provides a full version of the figure referenced in the main text (Figure 3). It expands the subset plot to all ten clusters and keeps axes, run counts, and error-bar conventions identical to the summary in Section 4.2. Use Figure 16 for detailed inspection of per-cluster behavior.

### G.3 Full Figure related to the CPT Experiment

Figure 17 shows how biased continued pre-training gradually increases preference for the exposed endpoint.

### G.4 Table related to the Reduction of Bias due to Mitigation

Table 5 demonstrates that our mitigation substantially flattens selection distributions and reduces both API- and position-level bias.

## H Effect of Metadata Perturbation on Bias

Relative to the base distributions, both models move farther from uniform (get more biased) when we lower semantic signal (see bars corresponding to the description + parameter and full

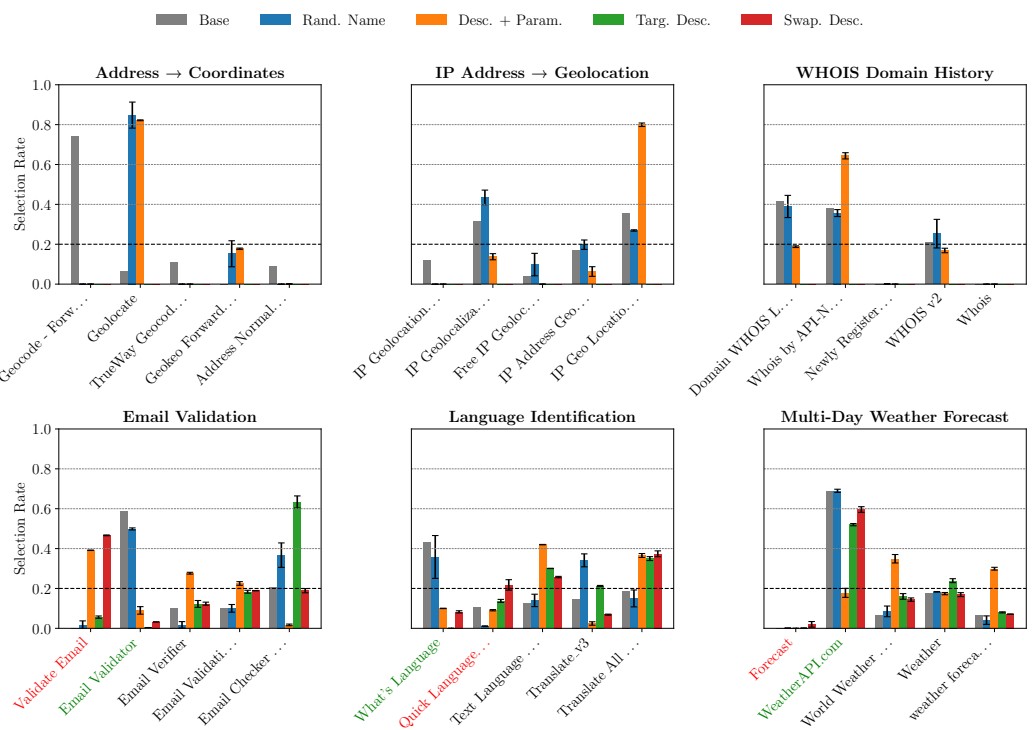

Figure 14: Selection distributions for Gemini under different name/ordering perturbations across six clusters of functionally equivalent APIs. Each subplot corresponds to one cluster; the x-axis lists the APIs and the y-axis shows the fraction of times the model under each condition selected that API, averaged over repeated runs. Error bars (when present) indicate the standard deviation across those repeats, making visible how robust or variable the preferences are under the different perturbations. Tools whose names are in green are the most selected by the baseline, and those in red are the least selected. These are the tools that are targeted for the swapping and selected scramble experiments.

Table 5: Average cluster-level API bias $\delta_{\mathrm{API}}$, positional bias $\delta_{\mathrm{pos}}$, and combined bias $\delta_{\mathrm{model}}$ before and after mitigation.

| Setup | $\delta_{\mathrm{API}}$ | $\delta_{\mathrm{pos}}$ | $\delta_{\mathrm{model}}$ |
|---|---|---|---|
| **Before** | 0.338 | 0.422 | 0.380 |
| **After** | 0.108 | 0.079 | 0.094 |

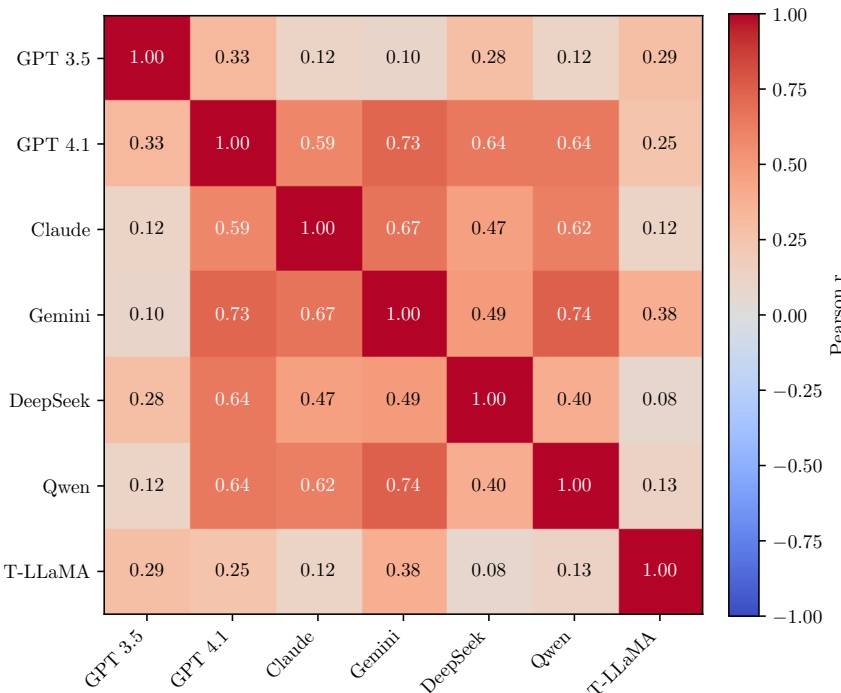

Figure 15: Pearson correlation matrix between models' tool-selection bias patterns.

perturbations in Figure 18). For Gemini, these manipulations yield the largest TV distances to uniform ($\approx$0.42–0.43); GPT shows similar results. This suggests that when descriptions/parameters are corrupted, models amplify bias rather than flatten choices.

Conversely, targeted edits to the most popular API tends to decrease bias. For Gemini, targeting the description of the most popular API leads to an average TVD slightly below the baseline and swapping the description between most- and least popular API leads to one that is substantially lower, indicating that weakening or transferring the strongest semantic cue moves the selection distribution toward uniform. Name-only manipulations have similar effects, but name scrambling does not increase bias as much.

## I  DETAILS ON CPT SETUP

**Model and adapters.** We continue pre-training Qwen3–8B–Base using Unsloth with 4-bit loading. The maximum sequence length is 2048. We attach LoRA adapters with ~16.29% trainable parameters. See Table 6 for more info.

Table 6: Model/adapter configuration.

| Base model | `unsloth/Qwen3-8B-Base-unsloth-bnb-4bit` |
|---|---|
| Max seq. length | 2048 |
| Quantization | 4-bit (bitsandbytes) |
| LoRA hyperparameters | $r = 128$, $\alpha = 32$, dropout$= 0$ |

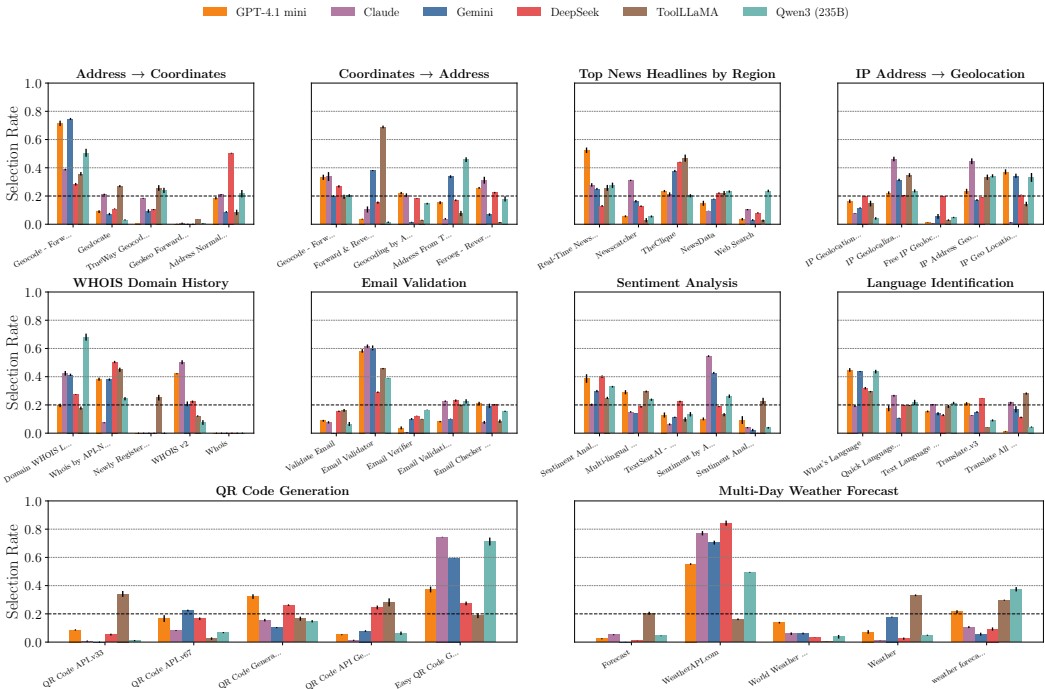

Figure 16: Selection distributions for six LLMs across ten clusters of functionally equivalent APIs. Each subplot corresponds to one cluster, with the x-axis indicating the API in the cluster and the y-axis showing the (mean) fraction of times each model chose that API over 500 inference runs; error bars indicate the standard deviation across three independent experimental runs. This visualization highlights how different models exhibit systematic preferences for some APIs.

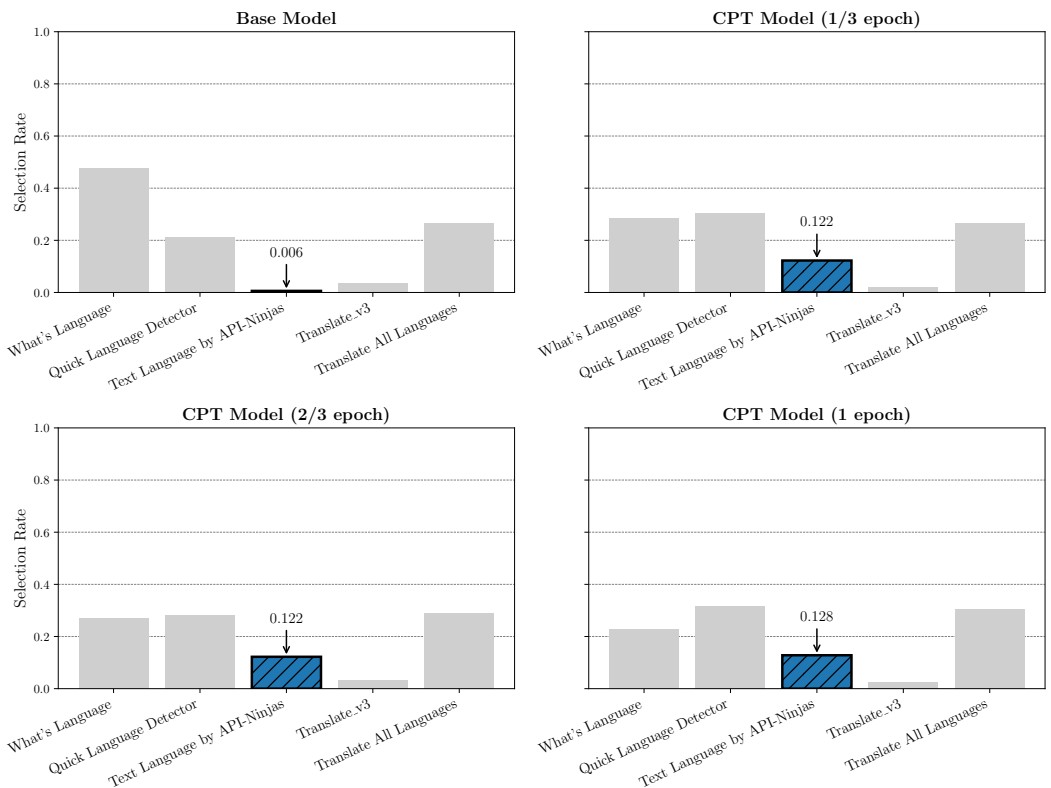

Figure 17: Selection rates for the Language Identification cluster across biased continued pre-training (CPT) checkpoints. Bars give the fraction that each endpoint was chosen across 500 inference runs. Panels show (top-left) base model, (top-right) CPT after 1/3 epoch, (bottom-left) 2/3 epoch, and (bottom-right) 1 full epoch. The Text Language by API-Ninjas endpoint is highlighted, with its exact selection rate printed above its bar. Differences across panels visualize how biased CPT shifts tool choice over training.

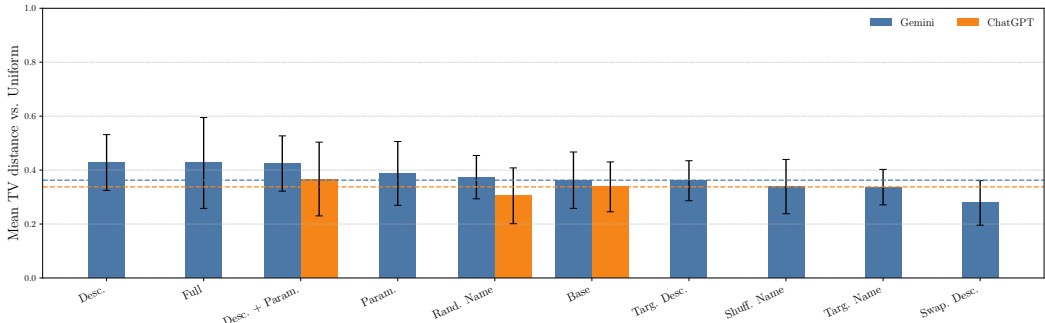

Figure 18: Mean total-variation (TV) distance from the uniform selection distribution to the distribution pertaining to each metadata perturbation (higher = more deviation from uniform). Blue bars show Gemini and orange bars show GPT; error bars denote standard deviation across clusters, and single bars indicate perturbations not run for GPT. Dashed horizontal lines (in the corresponding model colors) mark each model's baseline TV-to-uniform without perturbations.

**Data.** We use our corpus saturated with metadata of a single target endpoint (Text Language by API-Ninjas). The corpus contains ∼3.5M tokens.

**Training.** Training uses Unsloth's trainer for one epoch with cosine LR scheduler and warmup. Optimizer is 8-bit AdamW. We also set a smaller LR for the embedding modules. See Table 7 for more info.

Table 7: CPT training hyperparameters.

| | |
|---|---|
| Epochs | 1 |
| Total steps (epoch) | 153 |
| Per-device batch size | 2 |
| Grad. accumulation | 8 |
| Effective batch size | 16 |
| Learning rate | $5 \times 10^{-5}$ (embeddings $5 \times 10^{-6}$) |
| Scheduler / Warmup | cosine / warmup ratio 0.1 |
| Optimizer | `adamw_8bit` |
| Weight decay | 0.0 |
| Checkpoints used | step 0 (base), 52 ($\approx$1/3), 104 ($\approx$2/3), 153 (1 epoch) |

**Evaluation (inference).** For all checkpoints, we keep prompts and decoding fixed: temperature = 0.5, top-p = 1.0, and `max_new_tokens`=512. We evaluate with the Language Identification cluster under circular shifts, aggregating the selection rates over 500 inference runs per checkpoint.

## J EFFECT OF TOOL COUNT ON BIAS

To investigate how the number of available tools affects selection bias, we conducted an additional experiment using the *Sentiment Analysis* cluster (which contains five functionally equivalent APIs). From this cluster, we constructed subsets of size $K \in \{2, 3, 4\}$ using three selection strategies:

- **Best-to-worst:** select the $K$ APIs that were most frequently chosen in our initial $K = 5$ experiments.
- **Worst-to-best:** select the $K$ least frequently chosen APIs.
- **Random subsets:** uniformly sample $K$ tools from the cluster.

For each subset configuration, we re-ran the corresponding queries three times using Qwen3 (235B) and computed the normalized API-level and positional bias metrics.

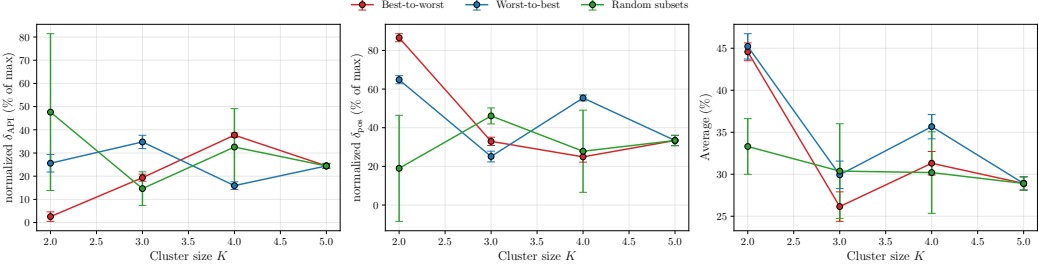

Figure 19: Normalized API-level and positional bias as a function of cluster size $K$ for the sentiment-analysis cluster. Each curve corresponds to one subset-selection strategy (best-to-worst, worst-to-best, random), with error bars indicating variability across three independent runs. Bias is reported as a percentage of its theoretical maximum $(1 - 1/K)$. The rightmost subplot shows the average of the normalized API-level and positional bias over $K$.

As can be seen in Figure 19, we find that bias is highly sensitive to the specific subset of tools shown to the model. At $K = 2$, both normalized $\delta_{\text{API}}$ and $\delta_{\text{pos}}$ vary substantially across selection strategies, revealing strong instability. At moderate subset sizes ($K = 3, 4$), the variance decreases and the measured bias becomes more stable, although not uniformly smaller. Overall, the relationship

between $K$ and bias is non-monotonic: depending on *which* tools are included, reducing the number of available tools can either amplify or attenuate bias. These findings indicate that the composition of the toolset is the primary driver of selection behavior, more than its size alone.

Finally, note that when $K = 5$ all methods trivially converge, since they expose the full cluster and therefore share identical toolsets. To fully characterize tool-count effects, future work should extend this analysis to larger clusters (e.g., $K = 6$–$10$).

## K    EXPLANATION OF LLM TOOL USAGE PIPELINE

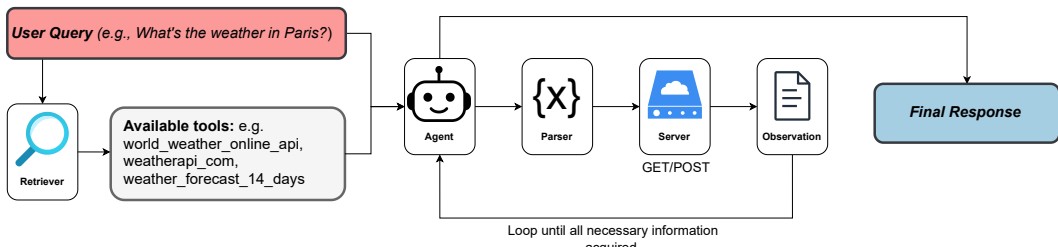

Figure 20: The overall workflow for tool usage with large language models. This figure illustrates the last three stages of tool usage: tool selection (via retriever and LLM selection), tool calling, and response generation (task planning is omitted). Note how the LLM's preliminary outputs need to be parsed to create a functional API call.

Large language models equipped with external tools typically follow a multi-stage decision and execution process. As summarized by prior work (Qu et al., 2025), the standard tool-usage pipeline consists of four stages: task planning, tool selection, tool calling, and response generation. Figure 20 showcases the latter three of these stages.

In the first stage, the LLM interprets the user query, identifies the user's intent, and (if necessary) decomposes it into sub-tasks that can be handled by external tools. Next, in the tool selection stage, the model determines which tools are suitable for each sub-task. Modern systems often employ a retriever-based filtering step before LLM-based selection: rather than presenting the descriptions of hundreds or thousands of available tools to the model (which is the case with large API hubs like RapidAPI), a retriever identifies the top-k most relevant candidates. This design is widely adopted in practical deployments due to context-length limitations (Qu et al., 2025).

In the tool calling stage, the LLM chooses one of the retrieved tools and generates the corresponding API call, including the required parameters in the correct format. This step requires the model not only to select an appropriate tool but also to extract and structure the tool arguments accurately. Once the LLM generation is parsed, the tool executes and returns its output, the LLM may decide to proceed to the final stage or determine that more tools need to be called. In the final stage, the LLM integrates the returned tool results into a coherent final answer to the user.

## L    CONSEQUENCES OF TOOL SELECTION BIAS

We have briefly touched on the consequences of tool selection bias in the introduction. In this section, we will further elaborate our points on why tool selection bias matters to give the reader a comprehensive view of the the effects and significance of the bias we uncover.

### L.1    ECONOMIC CONSEQUENCES

**Problem Statement.** A pay-per-request pricing model is common practice within tool marketplaces (see RapidAPI's pricing model or that of BridgeAPI). This means that usage of an API directly corresponds to the amount of revenue the developer of that API makes. Ideally, when two or more tools offer identical functionality, usage (and thus revenue) should also be split equally. This is not achieved when LLMs show consistent bias among functionality equivalent tools. This is not a

hypothetical concern. As we show in our experiments, some tools gets selected up **10 times** more than others whilst being functionally identical (see the geocoding cluster in Figure 16). If revenue is directly correlated with usage, the developer of this tool can then also expect a 10 times higher revenue than the developer of the disadvantaged tool for seemingly no particular reason but more advantageous metadata phrasing (see section 4.3 investigating why tool-selection bias occurs).

**Magnitude.** The size of the tool-usage market is hard to determine exactly. RapidAPI alone handles over 9 billion requests per month and recommends $0.00003 per API call *at a minimum*, a quick calculation gives us at least $270.000 in developer revenue per month or $3.24 million per year on this platform alone (RapidAPI (2025a)). Looking more broadly, the global API marketplace market size was estimated at $18.00 billion in 2024 and is projected to reach $49.45 billion by 2030, growing at a CAGR of 18.9% from 2025 to 2030 (GrandViewResearch (2025)). From this, one can see that even small shifts in automated traffic can have a meaningful economic impact. It's unclear how much of this API traffic is currently due to LLM agents, but with the increasing ubiquity of agents, we expect this share to rise significantly. This makes tool selection bias a tangible and pressing economic issue.

## L.2    USER EXPERIENCE

Tool-selection bias can directly degrade user experience when an LLM consistently favors an API that is objectively slower, less accurate, or more costly than its functionally equivalent alternatives. In such scenarios, end users may experience higher latency, lower-quality responses, or unnecessary costs, despite the existence of equally capable tools that would have delivered better performance. A more balanced usage distribution across equivalent APIs mitigates these issues by ensuring that no single suboptimal tool is disproportionately selected purely due to incidental metadata or positional biases. This leads to more stable, predictable, and higher-quality user outcomes.

## L.3    SAFETY & RELIABILITY

Biased tool selection magnifies the system's vulnerability to manipulated metadata and adversarial tools. Recent work on metadata-poisoning attacks (Mo et al. (2025)) shows that adversaries can craft strategically misleading tool names or descriptions to lure LLM agents into invoking harmful or unreliable APIs. When an LLM already exhibits strong, unintended preferences toward superficial metadata cues, such attacks become significantly easier to execute. In this sense, selection bias is not merely an efficiency problem. It increases the surface area for adversarial exploitation and directly undermines the reliability of agentic systems. A more uniform or semantics-invariant selection process would provide a stronger baseline defense by reducing the influence of manipulable metadata.

## L.4    EROSION OF TRUST & ECOSYSTEM EFFECTS

A further systemic consequence of tool-selection bias is the gradual erosion of trust in API marketplaces. If developers observe that LLM-mediated traffic does not meaningfully reflect the functional quality of their tools but instead hinges on arbitrary metadata preferences, they may perceive the marketplace as unfair or unpredictable. This creates incentives to bypass marketplaces entirely by hard-coding specific APIs into their applications. This outcome undermines the value proposition of marketplaces as neutral, competitive intermediaries. In the long term, this can reduce innovation, further distort competition, and create fragile ecosystems where a small number of arbitrarily preferred tools dominate traffic. Addressing tool-selection bias is therefore critical not only for individual user or developer outcomes, but for maintaining trust, participation, and healthy competition in the broader API economy.

