# OpenReview forum: "BiasBusters: Uncovering and Mitigating Tool Selection Bias in Large Language Models"
_ICLR.cc/2026/Conference — ICLR 2026 Poster_

### Official Review · Reviewer_xpw2 · 2025-10-26

**Soundness:** 4
**Presentation:** 3
**Contribution:** 3
**Rating:** 6
**Confidence:** 4

**Summary:**

This paper is the first work to study LLM tool-selection bias and provides a systematic evaluation across tasks and tools.
The authors design tasks that require invoking multiple candidate tools and measure how model choices vary across attributes and prompts. The core contributions of this paper include (1) a large-scale benchmark for measuring tool selection bias, (2) an in-depth analysis of the root cause of these biases, (3) a potential mitigation strategy.
Although this paper discusses an interesting and potentially impactful bias in the LLM era, the authors have provided a limited discussion of the impact of such a bias and the risks and implications behind it, which reduces the readability of this paper.

**Strengths:**

1. A large-scale benchmark for measuring tool selection bias
2. In-depth analysis of tool selection bias (including testing the impact of pre-training data on tool selection bias through training), which can strengthen people's understanding of this issue
3. Extensive experiments and analysis, providing valuable insights for future work on building fair and reliable agent systems

**Weaknesses:**

1. Lack of deep discussion of potential consequences of tool selection bias makes it difficult for readers to quickly get the effects and research significance of this paper.
2. Lack of a clear explanation of the LLM tool usage pipeline.

**Questions:**

1. Fig. 3 shows a long error bar in DeepSeek and the `what's language` question. The authors could provide a brief explanation and discussion of the results corresponding to this unusually large standard deviation, which would help readers better understand the model's biases and potential anomalous behavior.
2. Although this paper provides comprehensive experiments and analysis on the manifestations and sources of tool-selection bias, it lacks a discussion of the harmfulness and risks of this bias. The authors could supplement a discussion (for example, by providing figures to illustrate the value of the tool selection market) or conduct a questionnaire/interview to highlight the consequences of such a bias. This would further reinforce the significance and value of this work and echo the claim in the introduction and conclusion.
3. Since not everyone has a background in LLM tool usage, it will be good if the authors could provide a detailed tool usage process in the related work or appendix, preferably with a visual flowchart, which will effectively enhance the presentation of this paper.
4. Typo: There are no ChatGPT-3.5-turbo and ChatGPT 4.1-mini models. The authors may actually be referring to the GPT-3.5-Turbo and GPT 4.1-mini models.

---

> ### Author Response · Authors · 2025-11-25
> **Author Response**
>
> We thank the reviewer for the positive assessment, especially acknowledging (i) the novelty of the benchmark, (ii) the depth of our analyses, and (iii) the strength of the empirical results. We address the concerns and questions in detail below, and will incorporate the requested clarifications and discussions in the paper.
>
> **Weaknesses**:
> > Lack of deep discussion of potential consequences of tool selection bias makes it difficult for readers to quickly get the effects and research significance of this paper.
>
> We appreciate this suggestion. We have already done so briefly in the introduction. However, we will add a section dedicated to this discussion, which will contain the following points:
> - Economic consequences: how tool-selection bias can distort multi-provider tool marketplaces (e.g.,  Zhang et al. (2025) [1] show that code-generation LLMs disproportionately choose Google Cloud-style APIs over Azure/AWS equivalents, warning that such systematic favoritism can distort competition and contribute to market concentration and monopolistic dynamics). For large API marketplaces (e.g., RapidAPI, one of the world's largest, hosting ~80k APIs and ~7M developers and handling 9 billion calls per month), even small shifts in automated traffic can have a meaningful economic impact under pay-per-request pricing models
> - User experience consequences: biased tools can be slower, less accurate, or more expensive, affecting end users.
> - Safety and reliability risks: biased selection amplifies vulnerability to manipulated metadata and adversarial tools (connecting to recent work on metadata poisoning). Metadata-poisoning attacks (Mo et al., 2025 [2]) demonstrate that adversaries can craft misleading tool names/descriptions to lure LLM agents into selecting a malicious tool.
>
> [1]: The Invisible Hand: Unveiling Provider Bias in Large Language Models for Code Generation (Zhang et al., ACL 2025)
>
> [2]: Attractive Metadata Attack: Inducing LLM Agents to Invoke Malicious Tools (Mo et al, NeurIPS 2025)
>
> > Lack of a clear explanation of the LLM tool usage pipeline.
>
> We have added a figure (see Figure 20) and a text explaining the typical LLM tool usage pipeline (see the **newly-added** Appendix K). This should significantly improve readability for readers without prior background in LLM agent systems.
>
> **Questions**:
> > Fig. 3 shows a long error bar in DeepSeek and the what's language question. The authors could provide a brief explanation and discussion of the results corresponding to this unusually large standard deviation, which would help readers better understand the model's biases and potential anomalous behavior.
>
> Thank you very much for noticing this. On further inspection, DeepSeek seems to show quite different selection distributions for certain clusters when comparing different runs, explaining the high error bars. We suspect it could be the case that the third run (which is the most different) was run with a newer version of DeepSeek. We specified the usage of ‘deepseek-chat’ in our API call, which calls the latest version of DeepSeek; this could mean it had been automatically updated without us noticing between runs. We have run the DeepSeek experiments again, and the large error bars are gone now (see Figure 3 in the new paper).
>
> > Although this paper provides comprehensive experiments and analysis on the manifestations and sources of tool-selection bias, it lacks a discussion of the harmfulness and risks of this bias. The authors could supplement a discussion (for example, by providing figures to illustrate the value of the tool selection market) or conduct a questionnaire/interview to highlight the consequences of such a bias. This would further reinforce the significance and value of this work and echo the claim in the introduction and conclusion.
>
> We have added some discussion on general risks above. A questionnaire/interview study is an excellent idea, but outside the scope of this submission. We highlight it as future work.
>
> > Typo: There are no ChatGPT-3.5-turbo and ChatGPT 4.1-mini models. The authors may actually be referring to the GPT-3.5-Turbo and GPT 4.1-mini models.
>
> Thank you for catching this. We corrected the naming to “GPT-3.5-Turbo” and “GPT-4.1-mini” throughout the manuscript.

---

> > ### Comment · Reviewer_xpw2 · 2025-11-28
> >
> > Thanks for the authors' response. I have read the revised version, and the new content highlighted in red has solved most of my questions.
> >
> > There are still two minor issues:
> >  (1) I agree with the authors' solution of ​​adding a discussion. Considering space limitation, the authors could keep the original consequence introduction (the brief one) and provide a more detailed discussion in the appendix.
> >
> > (2) It would be best to mention the analysis and reproduction of the anomalous results from Deepseek in the paper, as this will help further improve the reliability and reproducibility of your experimental observation.
> >
> > I am willing to raise my score after solving these issues.

---

> > > ### Author Response · Authors · 2025-12-02
> > >
> > > We are glad to hear the new content has solved most of your questions. Please see the newly-revised paper incorporating 1) in the **newly-added** Appendix L and 2) in the reproducibility statement. We hope this solves the remaining issues you have highlighted. Thank you for pointing them out.

---

### Official Review · Reviewer_hsat · 2025-10-27

**Soundness:** 2
**Presentation:** 2
**Contribution:** 2
**Rating:** 4
**Confidence:** 4

**Summary:**

The paper exposes and addresses the problem of tool selection bias in LLM agent systems. It provides a benchmark, root cause analyses, and a simple bias mitigation approach. The results show bias with respect to an ideal scenario of uniform tool selection and the effectiveness of the mitigation as it directly achieves the target tool selection distribution by adopting a uniform distribution.

**Strengths:**

1. I appreciate the problem benchmarked by the work - tool selection bias. I think that this is an important problem.
2. The work is comprehensive - with a benchmark, exploratory root cause analyses, and a bias mitigation strategy.
3. The experimental setup is pretty comprehensive, covering multiple SOTA models and scenarios.

**Weaknesses:**

1. Methodology concerns:
    1. How does the LLM know that the tools are functionally equivalent when only metadata is exposed? From the LLM's perspective, the tools are not equivalent, and if the language of the metadata is more appealing, then it seems natural for the LLM to pick that. If instead of LLMs, we ask humans, then even their responses are not expected to produce uniform distribution over the tools. Hence, I think comparing with uniform distribution as the target doesn't seem appropriate. I would suggest comparing with some distribution learned from human responses, to make the LLMs respond similar to unbiased humans.
    2. To create the tool selection distributions from the LLM responses, I think it would be more appropriate to sample from LLM response multiple times.
    3. The method doesn't account for initial tool retriever biases, like [1], thus providing a partial picture of the tool selection bias.
    3. Other distance metrics, such as KL-divergence can be used between the LLM tool selection and uniform probability distributions.
    4. The proposed defense is a finer-grained retriever followed by an uninformed uniform distribution sampler. I wonder how it would perform compared to the initial retriever itself which produces a smaller slate of size K, directly followed by the uniform sampler.
    5. I think that RLHF preference training could be a more potential source for the tool selection bias than pretraining and would recommend studying that as well, alongside continual pretraining.
2. Results:
    1. Main results of the root cause analyses are expected and not novel. We can expect LLMs to exploit higher similarity between query and tool metadata, as they are after all doing pattern recognition.
    2. I recommend analyzing the chain of thought of models when the bias is high, to see if the bias might be intentional.
3. Related works: There are some missing comparisons to prior works, such as [2] (shows that LLMs can be manipulated to select specific adversarial tools) and [3] (shows LLM bias beyond benchmarking). Also, [1] goes beyond benchmarking for tool selection, and I wonder how a similar analysis can be used to expose tool selection biases more rigorously.

## References
1. Quantifying Distributional Robustness of Agentic Tool-Selection
2. From Allies to Adversaries: Manipulating LLM Tool-Calling through Adversarial Injection
3. Certifying Counterfactual Bias in LLMs

**Questions:**

1. What is your LLM decoding scheme to create the tool selection distribution?
2. What is the intuition behind "averaging" the API/tool and position probability distribution distances to form $\delta_{model}$?

---

> ### Author Response · Authors · 2025-11-25
> **Author Reponse 1/3**
>
> We thank the reviewer for their constructive and detailed feedback. We address each point below and provide clarifications, further experiments, and stronger justification for our design decisions.
>
> **Weaknesses**:
> > How does the LLM know that the tools are functionally equivalent when only metadata is exposed? From the LLM's perspective, the tools are not equivalent, and if the language of the metadata is more appealing, then it seems natural for the LLM to pick that.
>
> We made sure that each tool explicitly mentions the ability to perform the task given in each query, and the LLM should be able to pick up on that. The fact that, despite this, it still has a preference precisely captures the bias mechanism we aim to diagnose. The fact that superficial metadata leads to strong preferences, even when APIs and their functionalities are fully exchangeable, is evidence of undesirable sensitivity to name/description phrasing.
>
> We also clarify that our goal is not to assume LLMs choose between only functionally equivalent tools in practice. Instead, our setting models the scenario where a retriever has already surfaced a small set (e.g., 3–5) of tools that overlap in functionality. This is a situation that real systems increasingly produce as retrievers improve. Within such retrieved subsets, any functionally equivalent tools should receive equal opportunity, since systematic preference over equivalent options constitutes a selection bias (with potential fairness and financial implications, as discussed in the paper). Our experiments isolate this mechanism precisely.
>
> > If instead of LLMs, we ask humans, then even their responses are not expected to produce uniform distribution over the tools. Hence, I think comparing with uniform distribution as the target doesn't seem appropriate. I would suggest comparing with some distribution learned from human responses, to make the LLMs respond similar to unbiased humans.
>
> Our use of uniform selection as the ideal is not intended to model human preferences. Instead, it follows fairness definitions in multi-provider environments, where functionally equivalent providers should have equal opportunity in expectation unless there are genuine performance differences. This parallels fairness definitions in ranking systems and marketplaces (e.g., position-bias removal, exposure fairness). See, for example, Singh and Joachims (2018) [1], who formalize fairness in rankings as equal (or relevance-proportional) exposure for similarly qualified items. Treating uniform selection over functionally equivalent tools as the natural fairness benchmark in our setting follows naturally from this. Since all APIs in a cluster guarantee identical capability by construction (Sec. 3.2 on how it is constructed), any systematic deviation from uniformity is by definition unrelated to true utility, but reflects the bias inherited from model choices.
>
> That being said, we also agree that human-derived distributions can be insightful and could be future work. However, this is a different dimension in comparison to what we are doing.
>
> [1]: Fairness of Exposure in Rankings, Ashudeep Singh & Thorsten Joachims. Proceedings of the 24th ACM SIGKDD International Conference on Knowledge Discovery & Data Mining, 2018.
>
> > To create the tool selection distributions from the LLM responses, I think it would be more appropriate to sample from LLM response multiple times.
>
> We already do this: in all experiments, each query is run 5 times with cyclically rotated lists (Fig. 2). This design controls for decoding stochasticity and position effects much more rigorously than simply resampling. Also, during the runs, we have set the temperature to 0.5 (very low) to ensure minimum impact from sampling.
>
> > The method doesn't account for initial tool retriever biases, like [1], thus providing a partial picture of the tool selection bias.
>
> Correct, the goal of our work is to isolate the bias that occurs **after retrieval**. Retriever bias is important but orthogonal; combining the two would confound attribution and make understanding the root cause difficult. We would like to emphasise that our study intentionally fixes candidate lists to isolate behaviour within the LLM selection stage.
>
> > Other distance metrics, such as KL-divergence can be used between the LLM tool selection and uniform probability distributions.
>
> We actually computed KL in preliminary experiments. But found TV-distance is preferred because:
> - it is bounded,
> - directly interpretable as “fraction of mass that must be redistributed”, and
> - stable even when some selected probabilities are near zero.
>
> If the reviewer believes a comparison of distance metrics would be beneficial, we would be happy to expand the paper with an additional section, “On Divergence Metrics for Evaluating Bias”, to discuss KL, TV, and related alternatives for completeness.

---

> ### Author Response · Authors · 2025-11-25
> **Author Response 2/3**
>
> > The proposed defense is a finer-grained retriever followed by an uninformed uniform distribution sampler. I wonder how it would perform compared to the initial retriever itself which produces a smaller slate of size K, directly followed by the uniform sampler.
>
> We appreciate the reviewer’s suggestion. Whilst an interesting idea, the reason we do not directly compare against a retriever → uniform sampling baseline using the original retriever output (i.e., without refinement) is that practical retrievers typically have imperfect precision. Uniform sampling over such a subset would frequently select tools that cannot perform the user’s task. Our mitigation pipeline, therefore, first increases precision by refining the subset (see Sec. 4.4), and only then applies uniform sampling.
>
> More broadly, our goal is to study the origins of the bias itself. Understanding positional and semantic preference is the main contribution of this work; comparing defenses is a slightly orthogonal direction, but we agree is promising for future research.
>
> > I think that RLHF preference training could be a more potential source for the tool selection bias than pretraining and would recommend studying that as well, alongside continual pretraining.
>
> We appreciate the reviewer’s suggestion. We agree that RLHF can introduce general preference patterns into LLMs. However, we note that current tool-capable LLMs are not typically RLHF-trained on tool selection itself. To the best of our knowledge, there is no large-scale human preference dataset comparing equivalent tools for the same task. In practice, tool-use abilities are predominantly learned through supervised fine-tuning on synthetic or curated tool-call traces, not RLHF (see the training of ToolLLaMA in [2]).
> For this reason, while RLHF may contribute indirectly to general linguistic or stylistic biases, we believe it is unlikely to be the primary mechanism behind the tool-selection-specific biases we study. Instead, our controlled continual-pretraining study (Sec. 4.3) demonstrates that semantic exposure during training is sufficient to induce tool preferences, even in the absence of RLHF.
>
> That said, we agree that isolating RLHF contributions would be valuable future work (as it could bias the model towards the writing styles of certain tool descriptions/names), and we will note this in the discussion section.
>
> [2]: Yujia Qin, Shihao Liang, Yining Ye, Kunlun Zhu, Lan Yan, Yaxi Lu, Yankai Lin, Xin Cong, Xiangru Tang, Bill Qian, Sihan Zhao, Lauren Hong, Runchu Tian, Ruobing Xie, Jie Zhou, Mark Gerstein, Dahai Li, Zhiyuan Liu, and Maosong Sun. Toolllm: Facilitating large language models to master 16000+ real-world apis. In The Twelfth International Conference on Learning Representations, ICLR, 2024.
>
> > Main results of the root cause analyses are expected and not novel. We can expect LLMs to exploit higher similarity between query and tool metadata, as they are after all doing pattern recognition.
>
> While semantic alignment seems intuitive, our contribution is that:
> 1. We quantify its effect, showing it is the dominant factor but only explains <40% of the variance (R^2 < 0.4 for all models, dipping as low as 0.12 for Gemini). Therefore, alignment does, by no means, paint a complete picture of the bias.
> 2. The perturbation experiments reveal strong, non-intuitive behaviours, such as:
> - selective inversion of preferences under targeted description corruption,
> - strong bias even when semantics are corrupted.
> 3. No prior work systematically evaluates bias in non-adversarial, functionally interchangeable tool settings.
>
> We also note that conducting a rigorous, large-scale study on intuitively suspected behaviour is well-established in top-tier venues. For example, [3] analyses tokenizer-induced unfairness across languages, which was long suspected but never systematically quantified until this work. Our study follows the same tradition: turning an intuitive concern into a concrete, measurable, reproducible analysis.
>
> [3]: Aleksandar Petrov et al. “Language Model Tokenizers Introduce Performance Disparities Across Languages.” NeurIPS 2023.
>
> > I recommend analyzing the chain of thought of models when the bias is high, to see if the bias might be intentional.
>
> We thank the reviewer for the suggestion and agree that CoT analysis could be valuable. We now allow the model to reason for one step before picking a tool to use. From a preliminary inspection of the logs, we note that LLMs often do not give an exact reasoning for the selection of any particular tool. However, if we allowed the LLM to reason for longer, it would perhaps give a better explanation for the tool it chooses. Due to computational limitations, we leave this for future work, and we now explicitly mention it in our discussion and future work section.

---

> ### Author Response · Authors · 2025-11-25
> **Author Response 3/3**
>
> > Related works: There are some missing comparisons to prior works, such as [2] (shows that LLMs can be manipulated to select specific adversarial tools) and [3] (shows LLM bias beyond benchmarking). Also, [1] goes beyond benchmarking for tool selection, and I wonder how a similar analysis can be used to expose tool selection biases more rigorously.
>
> We thank the reviewer for highlighting [1–3]. We have added them and clarified differences:
> 1. [1] and [2] focus on tool selection attack rather than natural bias in the selection process.
> 2. [3] studies specifically bias from demographic groups, instead of functionality, which is not directly related to our research problem. Once again in [3], the bias is not natural but rather injected via the prompt, which significantly differs from our research here.
>
> After looking at the literature, we were further convinced that we are doing something really novel, and we thank the reviewer for contributing to the related work for our paper. Our related work section will definitely benefit from these additions, and we plan on adding them.
>
> **Questions**:
> > What is your LLM decoding scheme to create the tool selection distribution?
>
> This is mentioned in the paper. Temperature 0.5, top-p 1.0, 5 samples per rotated list (Sec. 4.1).
>
> > Why average API-level and positional distances?
>
> They capture orthogonal axes of bias:
> 1. API-level: true preference independent of list position
> 2. Positional: structural bias due to ordering
>
> Averaging avoids overweighting either component.

---

### Official Review · Reviewer_kirZ · 2025-10-29

**Soundness:** 2
**Presentation:** 3
**Contribution:** 2
**Rating:** 4
**Confidence:** 5

**Summary:**

This paper investigates the causes behind tool selection biases, where LLMs exhibit biases towards certain tools based on their descriptions or positions in the prompt even when the tools are equivalent in terms of their functionality. The authors also propose a simple mitigation strategy to alleviate tool selection biases, and empirically show that it can reduce tool selection bias.

**Strengths:**

1. In general, LLMs' selection bias is an important yet unsolved challenge despite extensively studied in recent literature. Within this broader context, tool selection bias is a less studied subproblem that worth further investigations. This work aims to address this gap.

2. Measuring bias of API itself (δ_API) and positional bias (δpos) separately is reasonable and disentangle the biases arising from these two different sources.

3. The idea behind the mitigation strategy, which recognizes the subset of tools that can address the given query and then performs uniform sampling from the subset, is intuitive and easy to implement.

4. The authors evaluate a variety of LLMs and API endpoints.

**Weaknesses:**

1. (Informativeness of the findings) Selection biases and their mitigation strategies have been studied extensively in previous literature. For example, positional biases are well-known in the research community (see the "missing references" below). While I acknowledge that there are special issues unique to "tool" selection bias, such as tool descriptions and metadata, I think the contributions of this work are relatively limited considering the presence of existing works.

2. (Missing references of selection bias) Following the previous points, I believe these highly relevant yet missing references need to be cited in this paper:

[1] Pezeshkpour, Pouya, and Estevam Hruschka. "Large language models sensitivity to the order of options in multiple-choice questions." arXiv preprint arXiv:2308.11483 (2023). NAACL 2024 Findings

[2] Zheng, Chujie, et al. "Large language models are not robust multiple choice selectors." arXiv preprint arXiv:2309.03882 (2023). ICLR 2024 Spotlight

[3] Wei, Sheng-Lun, et al. "Unveiling selection biases: Exploring order and token sensitivity in large language models." arXiv preprint arXiv:2406.03009 (2024). ACL 2024 Findings

[4] Wang, Ziqi, et al. "Eliminating position bias of language models: A mechanistic approach." arXiv preprint arXiv:2407.01100 (2024). ICLR 2025 Poster

**Questions:**

1. As selection bias is a well-studied issue of LLMs. What are the special issues in tool selection bias compared to selection bias in general that require a dedicated research? Elaborate more on this can strengthen the positioning of this paper.

2. Are the bias due to the fact that the tools are simply better described and understandable by LLMs? From your experiment results that "semantic similarity between queries and tool descriptions is the strongest predictor of selection", another way to interpret that is the descriptions are better aligned with the user query, rather than interpreting it as "bias". I would like to hear the authors' comments on this.

3. I also have a fundamental question about the setting in this paper. From a practical point of view, if the APIs are equivalent in their functions ("functionally interchangeable" as descsribed in line 160), we don't have to include all of them in the prompt every single time. If practitioners need to consider fairness or balance usage, they just have to uniformly sample a tool from the equivalent tools of each functional cluster and put them into the prompt for LLMs to choose from. Therefore, I am curious about how important and realistic the setting is.

---

> ### Author Response · Authors · 2025-11-25
> **Author Response 1/3**
>
> Thank you for your review. We appreciate your suggestions for improving the paper, and we hope that the following response will help to alleviate any of your concerns about parts of the paper that were unclear. Please find our responses to the weaknesses and questions below:
>
> **Weaknesses**:
> > Informativeness of the findings) Selection biases and their mitigation strategies have been studied extensively in previous literature. For example, positional biases are well-known in the research community (see the "missing references" below). While I acknowledge that there are special issues unique to "tool" selection bias, such as tool descriptions and metadata, I think the contributions of this work are relatively limited considering the presence of existing works.
>
> We will update the related work section with a section on positional biases on LLMs using these references. Thank you. That being said, we agree that positional biases have been shown before in different contexts. However, demonstrating that this behaviour persists for tool selection is just one finding of our work. When controlling for this bias, we demonstrate in our experiments that other, more complex biases persist and show just how persistent this bias is (remaining even, for example, when all semantics from descriptions and tool names are removed). Moreover, we also note that due to the different context and financial incentives involved with API selection (in some marketplaces APIs earn money per request[1][2]), positional biases are a more pressing issue here.  Unfair preferential positioning by LLMs can translate into significant economic disparities (potentially thousands or millions of dollars in diverted revenue), making fairness in tool selection a particularly urgent issue.
>
> > (Missing references of selection bias) Following the previous points, I believe these highly relevant yet missing references need to be cited in this paper:
>
> As indicated above, we will endeavor to update the paper with these missing references. Again, we’d like to note our analysis investigates many other biases over and above positional biases.
>
> [1]: BridgeAPI. Pricing. BridgeAPI. https://www.bridgeapi.store/marketing/pricing. Accessed 24th of November, 2025.
>
> [2]: RapidAPI. Monetizing your API on RapidAPI. RapidAPI Documentation. https://docs.rapidapi.com/docs/monetizing-your-api-on-rapidapicom. Accessed 24th of November 2025.

---

> ### Author Response · Authors · 2025-11-25
> **Author Response 2/3**
>
> **Questions**:
> > As selection bias is a well-studied issue of LLMs. What are the special issues in tool selection bias compared to selection bias in general that require a dedicated research? Elaborate more on this can strengthen the positioning of this paper.
>
> The unique issues of tool selection bias come from a number of sources:
> 1. To the best of our knowledge, most previous studies have considered selection problems where the options provided have one “correct” option, such as the papers on multiple-choice questions you have referenced. Conversely, the problem of tool selection is selecting a valid option from a set of options where some subset might be valid, and the rest aren’t. In our paper, to simplify the analysis, we focus on the case where all options are valid; hence, we study the bias when selecting between functionally equivalent options that are all correct. The question we try to answer here is if models would still prefer some options over others despite the fact that all tools are equally capable. This is pressing because of the financial factors involved and the possibility of unfairness across API/tool developers. That being said, a follow-up benchmark where not all APIs could fulfill all requests would be an interesting albeit different direction for future work.
> 2. API providers have a financial incentive to manipulate their tool descriptions so their tools are selected more frequently. Indeed, recent work shows that adversarial actors (or tool-providers) can manipulate tool metadata to steer LLM agents toward specific tools despite functional equivalence [3][4]. This is not the case in most previous contexts where selection biases have been investigated, in which the “answers” are static and thus not open to manipulation. This demonstrates that the selection bias we study is not only inherent but also exploitable, thereby creating risks of unfair competition and revenue diversion. Where in other contexts, while biases are still undesirable, it is unlikely the environment will change to make them more pronounced.
> 3. Context matters - While selection bias has been studied in multiple question answering which share a structure with tool selection, the context, vocabulary, and subset of the natural language and training data are different. There are no guarantees that the behaviour would be similar here. Given the financial incentives, we believe this area was in need of dedicated research.
> 4. Lastly, LLM agents utilizing tools are seeing rapidly increasing real-life deployment. We believe our research sheds some light on how one could achieve fair, unmanipulable agent deployment, avoiding a plethora of issues in the process.
>
> [3]: Kanghua Mo, Li Hu, Yucheng Long, and Zhihao Li. Attractive metadata attack: Inducing llm agents
> to invoke malicious tools. arXiv preprint arXiv:2508.02110, 2025.
>
> [4]: Kazem Faghih, Wenxiao Wang, Yize Cheng, Siddhant Bharti, Gaurang Sriramanan, Sriram Bala-
> subramanian, Parsa Hosseini, and Soheil Feizi. Gaming tool preferences in agentic llms. arXiv
> preprint arXiv:2505.18135, 2025.

---

> ### Author Response · Authors · 2025-11-25
> **Author Response 3/3**
>
> > Are the bias due to the fact that the tools are simply better described and understandable by LLMs? From your experiment results that "semantic similarity between queries and tool descriptions is the strongest predictor of selection", another way to interpret that is the descriptions are better aligned with the user query, rather than interpreting it as "bias". I would like to hear the authors' comments on this.
>
> It’s indeed the case that one of our findings is that better alignment between tool descriptions and user queries leads to more frequent selection (as discussed in section 4.3). Whether this phenomenon is seen as a useful property or an unwanted “bias” likely depends on the setting being considered. In settings with unequivalent APIs, selecting the API whose description is best aligned with the query might not be problematic, as it will likely be the best API to use. However, this does not hold when we have multiple APIs that are functionally equivalent, which means they can all perform the user-specified task to the same level. This setting is representative of the usual tool-usage agent setting; often, multiple APIs can fulfil the user request. For a concrete example, RapidAPI has tens of thousands of APIs in their database, with multiple APIs that can perform the same task[5]. This is the setting we investigate in our experiments. In this setting, this phenomenon is negative as it can lead to unfair selection of APIs and unwanted incentive structures.
>
> For example, selection biases mean that a new API can be added to a database with a specially designed description that would now be preferred over existing tools for certain requests, even though the existing tools were perfectly able to complete these requests, were previously used for these requests, and had a description making this clear. This creates an incentive structure for API providers to optimise tool descriptions in order to maximise usage and hence profits. We claim this is not useful competition as the service of the APIs is in no way being improved, just their “marketing”. This is a similar phenomenon to search engine optimization. This is not a hypothetical concern either; there are already works investigating such manipulations [4]. However, we would like to make clear that our paper investigates what causes these biases in real-world APIs, rather than how these insights can be exploited for maximal gain, as knowing the root causes can help us understand attack surfaces for adversaries aiming to manipulate this unfairly.
>
> Lastly, the mentioned alignment is indeed the strongest feature of our predictor, but it must be noted that our predictor is quite weak (R^2 < 0.4 for all models, dipping as low as 0.12 for Gemini). Therefore, alignment does, by no means, paint a complete picture of the bias.
>
> > I also have a fundamental question about the setting in this paper. From a practical point of view, if the APIs are equivalent in their functions ("functionally interchangeable" as descsribed in line 160), we don't have to include all of them in the prompt every single time. If practitioners need to consider fairness or balance usage, they just have to uniformly sample a tool from the equivalent tools of each functional cluster and put them into the prompt for LLMs to choose from. Therefore, I am curious about how important and realistic the setting is.
>
> This would indeed be the case if one had access to a neat subset of APIs that are all equivalently effective for each possible user query. It is important to remember that, unfortunately, in general, this is not the case. In a realistic setting, a tool retrieval will fetch a set of tools from the full database of tools in line with the user query. Uniformly sampling from a retrieval system that does not have perfect precision and recall will lead to unwanted behaviours. A non-perfect precision leads to APIs being queried that cannot fulfill the request. A retriever with non-perfect recall leads to some valid APIs not getting selected at all. As all retrieval systems have imperfect precision and recall, we will run into these issues. In the paper, we therefore construct perfect functionally equivalent API lists by hand, but in general, you have no guarantee that any retrieval system will provide a list like this. Once you have a valid subset of tools/APIs that can fulfil a request, uniform sampling is indeed a good way of removing bias. In Section 4.4, we propose a method of removing bias by first constructing a list of valid APIs, which we then subsample from. To the best of our knowledge, this is the first time a method like this has been proposed for removing bias in tool selection. Hopefully, this answers your question.
>
> [5]: RapidAPI. Home Page. RapidAPI. https://rapidapi.com/. Accessed 24th of November, 2025.

---

### Official Review · Reviewer_CTUZ · 2025-10-31

**Soundness:** 4
**Presentation:** 3
**Contribution:** 4
**Rating:** 8
**Confidence:** 4

**Summary:**

This paper studies tool-selection bias in tool-augmented LLMs, showing that models often favor certain APIs for reasons unrelated to utility. The authors build a benchmark of functionally equivalent tools, measure bias across major LLMs, and find semantic metadata and training exposure as key drivers. They further demonstrate that controlled metadata changes shift model choices and propose a simple mitigation (i.e., filter relevant tools then uniformly sample), that effectively reduces bias without harming performance.

**Strengths:**

1, This paper studies tool selection bias in LLMs, which is a very new and meaningful topic.

2, The authors build a clear benchmark with multiple tool clusters and balanced queries. The experiments are systematic across several strong LLMs, and the evaluation metrics are reasonable.

**Weaknesses:**

1, The paper does not probe whether LLMs' own familiarity/knowledge about each API explains its selection bias. Current analyses focus on metadata features, perturbations, and exposure effects, but never explored the model’s prior knowledge of each candidate API, so it’s unclear whether choices reflect surface cues or genuine familiarity.

2, The paper fixes the number of candidate APIs, leaving unexplored how bias scales or changes when the number of tools varies (e.g., 2,3,4...). This is important to understand whether larger API pools amplify or dilute preference concentration is important for real-world marketplaces where tool pool size varies.

**Questions:**

N/A

---

> ### Author Response · Authors · 2025-11-25
> **Author Response**
>
> Thank you for your kind words about our paper, especially highlighting that you think it addresses a very new and meaningful topic.
>
> Please find our responses to the weaknesses you pointed out below:
> > The paper does not probe whether LLMs' own familiarity/knowledge about each API explains its selection bias. Current analyses focus on metadata features, perturbations, and exposure effects, but never explored the model’s prior knowledge of each candidate API, so it’s unclear whether choices reflect surface cues or genuine familiarity.
>
> This is already partly answered in our paper. In Section 4.3, under “Does additional pre-training exposure to one endpoint change selection distribution?”, we show that increasing familiarity/knowledge of an LLM to an API does have an effect on its bias. We show that a model fine-tuned on data that frequently mentions a specific API and its functionality also calls it with an increased frequency. This shows that increased prior knowledge of a candidate API does, indeed, influence the selection. However, we agree that a similar experiment should be run with larger LLMs, but this is outside our computing capabilities. Additionally, in Figure 5 and Section 4.3, we show scrambling API names, which would remove any knowledge of specific APIs outside of the description, has a limited effect on the bias; however, its effect can vary from setting to setting. Quantifying an LLM's own familiarity/knowledge about an API in a more accurate way is surprisingly difficult. We agree that applying mechanistic interpretability techniques or training LLMs from scratch with controlled training data to better quantify this would be an interesting direction for future work.
>
> > The paper fixes the number of candidate APIs, leaving unexplored how bias scales or changes when the number of tools varies (e.g., 2,3,4...). This is important to understand whether larger API pools amplify or dilute preference concentration is important for real-world marketplaces where tool pool size varies.
>
> We agree that this is an interesting ablation study, and following the reviewer's suggestion, we run an experiment varying the number of tools in context, which is described in the **newly-added** Appendix J. For this experiment, we picked the sentiment analysis cluster and subsampled 2, 3, or 4 tools from it. We utilized three methods for picking the tools:
> - Best-to-worst: select the K APIs that were most frequently chosen in our initial K=5 experiments.
> - Worst-to-best: select the K least frequently chosen APIs.
> - Random subsets: we randomly pick K tools from the total cluster.
>
> We then ran through the corresponding queries again three times using Qwen3 (235B) and plotted our bias metrics for each cluster size (normalized as the max bias value is dependent on K). As can be seen in **Figure 19**, we observe that bias is highly sensitive to the specific subset presented to the model. At K=2, both API-level and positional bias vary dramatically across selection methods, indicating strong instability. As can be seen in the middle plot, positional bias is especially high when using the non-random methods (60-80%). This could be explained by the fact that the two tools in context had similar selection rates in the initial K=5 experiment and have therefore similar semantic ‘strength’. The LLM, having no strong preference between the two, falls back on positional heuristics. At moderate subset sizes (K=3,4), the variance decreases, and bias becomes more predictable. Overall, the effect of the number of tools in context is non-monotonic: depending on which tools are included, reducing K can either increase or decrease bias. This suggests that the composition, not merely the size, of the available toolset is the primary determinant of tool-selection bias. Investigating whether these trends continue for K>5 would be an interesting direction for future work. To the best of our knowledge, K = 5 and 10 are popular choices. For example, ToolLLM [1] uses K=5 [1], and Shi et al. (ACL 2025) [2] use K=10.
>
>
> [1]: Yujia Qin, Shihao Liang, Yining Ye, Kunlun Zhu, Lan Yan, Yaxi Lu, Yankai Lin, Xin Cong, Xiangru
> Tang, Bill Qian, Sihan Zhao, Lauren Hong, Runchu Tian, Ruobing Xie, Jie Zhou, Mark Gerstein,
> Dahai Li, Zhiyuan Liu, and Maosong Sun. Toolllm: Facilitating large language models to master
> 16000+ real-world apis. In The Twelfth International Conference on Learning Representations,
> ICLR, 2024.
>
> [2]: Zhengliang Shi, Yuhan Wang, Lingyong Yan, Pengjie Ren, Shuaiqiang Wang, Dawei Yin, and Zhaochun Ren. 2025. Retrieval Models Aren’t Tool-Savvy: Benchmarking Tool Retrieval for Large Language Models. In Findings of the Association for Computational Linguistics: ACL 2025, pages 24497–24524, Vienna, Austria. Association for Computational Linguistics.

---

> > ### Comment · Reviewer_CTUZ · 2025-11-25
> > **Author rebuttal acknowledged**
> >
> > Author rebuttal acknowledged

---

### Meta-Review · Area_Chair_RG6j · 2025-12-25

**Summary:**

The paper studies the LLM selection bias in the context of tool usage, revealing that LLMs exhibit bias in favour of specific choices thus deviate uniform tool-selection. The authors build a benchmark with 1000 queries spanning 10 tasks (with 5 equivalently functioning APIs for each task) to evaluate the tool selection bias, conduct further analysis (e.g., linear regression) to explore factors that cause the bias, and propose a filter-then-uniform-sample strategy to mitigate the bias issue. Reviewers’ concerns include the special issues of this tool-use selection bias (reviewer kirZ, xpw2), limited informativeness of findings (reviewer kirZ, hsat), and reasonability of setting (kirZ). Considering the overall quality of my batch, I still weigh in for acceptance. Hope the authors can address the concerns of the above concerns in the final version.

**Reviewer Concerns:**

- Reviewer CTUZ raised concerns that LLMs' internal knowledge about the APIs is not studied, and the number of candidate APIs is fixed. The authors defended by further articulating existing experiments and adding additional experiments.
- Reviewer kirZ's main concerns include the limited informativeness of findings, special issues regarding tool-use selection bias remain unclear, the unclear root of bias, missing related works about LLM selection bias, and the necessity of fundamental settings. During the rebuttal, the authors distinguish tool-use selection bias from a simple adaptation of positional bias in the context of tool-use case (this is validated by the bias disentanglement in experiments); however, I remain unconvinced about the practical impact of this tool-selection bias and the contribution of the proposed mitigation strategy, and there still remains a question about how to justify a smaller LLM can return valid tool lists while the tool-augmented LLM cannot?
- Reviewer hsat raised concerns about the equivalence between API candidates, robustness of the sampling process, and missing related works, which are addressed during rebuttal. It is also noted that the finding aligns with expectations that semantic similarity plays a significant role in tool selection, which the author claimed that it does not paint the whole picture for the tool selection bias.
- Reviewer xpw2 is mainly concerned about the potential and practical consequences of tool-use selection bias. While the authors provided discussion, I remain unconvinced by the practicality of this issue, e.g., from the perspective of user experience, this found bias might have either positive or negative impact depending on which specific API is favoured.

**Reviewer Scores:**

Reviewer kirZ and hsat would tend to keep initial scores and CTUZ and xpw2 would keep positive.

---

### Decision · Program_Chairs · 2026-01-26

Accept (Poster)